# Online Facility Location with Multiple Advice

**Matteo Almanza**
Dipartimento di Informatica
Sapienza University
Rome, Italy
almanza@di.uniroma1.it

**Flavio Chierichetti**
Dipartimento di Informatica
Sapienza University
Rome, Italy
flavio@di.uniroma1.it

**Silvio Lattanzi**
Google Research
Zurich, Switzerland
silviol@google.com

**Alessandro Panconesi**
Dipartimento di Informatica
Sapienza University
Rome, Italy
ale@di.uniroma1.it

**Giuseppe Re**
Dipartimento di Informatica
Sapienza University
Rome, Italy
re@di.uniroma1.it

## Abstract

Clustering is a central topic in unsupervised learning and its online formulation has received a lot of attention in recent years. In this paper, we study the classic facility location problem in the presence of multiple machine-learned advice. We design an algorithm with provable performance guarantees such that, if the advice is good, it outperforms the best-known online algorithms for the problem, and if it is bad it still matches their performance. We complement our theoretical analysis with an in-depth study of the performance of our algorithm, showing its effectiveness on synthetic and real-world data sets.

## 1   Introduction

Clustering is a central topic in unsupervised learning [Jain and Dubes, 1988, Aggarwal and Reddy, 2014, Gan et al., 2020] In the past few years, its online version has gained a lot of attention [Zhang et al., 2004, Liberty et al., 2016, Lattanzi and Vassilvitskii, 2017, Cohen-Addad et al., 2021]. In this paper we are interested in the fundamental question of whether machine-learned advice, especially when coming from multiple sources, can boost the performance of clustering algorithms in the online setting. In particular, we are interested in designing online algorithms with access to multiple advice that exhibit strong theoretical guarantees together with a good experimental performance.

To explore the interplay between online algorithms and multiple advice we focus on facility location, a classic clustering problem with a long and rich history in computer science and operations research [Korte and Vygen, 2018], which can also be seen as the Lagrangian relaxation of $k-$median clustering. In the online setting of $k-$median clustering, it is easy to see that if the requirement of having $k$ clusters is a hard constraint then the cost of solutions is bound to be arbitrarily far from the optimum, which makes the straightforward online formulation rather uninteresting algorithmically. There are two standard ways in which the constraint can be relaxed: bi-criteria solutions and the Lagrangian relaxation. Facility location captures the latter.

In the online version of facility location, points arrive in sequence and, as soon as a point arrives, an irrevocable decision must be made: we either assign it to an existing opened facility or we open a new one to serve it. Each facility, therefore, induces a cluster, consisting of the points assigned to it. The cost of this sequence of decisions is given by the sum of the service costs (*i.e.* the distance between a point and its facility) and the total facility cost (opening each facility has a cost, which in this paper we assume to be uniform). The goal is to find as cheap a clustering as possible. This online version

was first introduced by Meyerson [2001], who gave an elegant $O(\log(n))$-approximation algorithm, and studied extensively thereafter [Fotakis, 2003, 2005, 2011, Cygan et al., 2018, Cohen-Addad et al., 2019, Ahmed et al., 2020, Guo et al., 2020]. One of the main reasons for this heightened attention is that even in dynamic settings the resulting clustering is very stable. This is particularly useful in real-world systems where clusters are served directly to users or when they are used in a downstream machine learning model, a situation where modifying the clustering would incur too high a cost.

In real-world applications, one typically has side information that could help solve the online clustering problem. This natural scenario is a recurrent theme in machine learning that focuses on how to take advantage of expert advice. In the context of online algorithms, Lykouris and Vassilvitskii [2018] and Mahdian et al. [2012] introduced a formal framework for online problems within which mathematically rigorous performance guarantees can be given in terms of the quality of the advice. In particular, they showed how to incorporate the advice in a *robust* way: the advice is exploited when it is good and disregarded when it is bad. All this happens automatically: the robust algorithm does not need to know in advance which is which. Remarkably, when the advice is bad the performance of the best online algorithm is matched. The approach combines mathematical rigor with practical effectiveness, adding the desirable dimension of robustness. The solution we develop for facility location is in the same spirit, with a focus on multiple advice which adds an important new dimension. Thus, the goal of this paper is to show how to take advantage of multiple advice within this rigorous algorithmic framework in the case of online facility location.

In our set-up the advice comes in the form of a family of sets, each suggesting a list of facilities to open. As we show with our experiments, multiple advice of this form can be easily and efficiently produced on the basis of past data. The main technical result of this paper is the following. Suppose we have a family of sets $\mathcal{S}_1, \ldots, \mathcal{S}_k$ which intuitively represent sets of suggested facilities to open, and let $\mathcal{S}$ be their union. We develop an online randomized algorithm, called TAKEHEED, that computes a solution whose expected cost is

$$\mathcal{O}\left(\log(|\mathcal{S}|) \cdot \text{OPT}(\mathcal{S})\right),$$

where $\text{OPT}(\mathcal{S})$ is the best solution that can be obtained using only facilities in $\mathcal{S}$. Notice that the $\text{OPT}(\mathcal{S})$ factor is no worse than any of the suggested sets in isolation since $\text{OPT}(\mathcal{S}) \leq \min_i \text{OPT}(\mathcal{S}_i)$. In fact, it can be much better. For instance, every set could contain just a few good points and a lot of noise. Or, one of the sets could be good but its identity unknown. And yet, TAKEHEED separates the wheat from the chaff and delivers a near-optimal solution. This result (which is the main technical step toward the solution) can be seen as a solution to the version of facility location in which the facilities are constrained to be in a specific set, as opposed to the entire input metric space. The above result is essentially the best possible. In Section 4, we prove that the $\log |\mathcal{S}|$ factor is near-optimal. We also show that the best-known algorithms (see Meyerson [2001], Fotakis [2003, 2005]) cannot match the same bound, for they can only attain an $\Omega(|\mathcal{S}|)$ approximation factor in general. This is a strong indication that techniques different from those employed by those state-of-the-art algorithms are required for this problem, providing a compelling motivation for our approach which is based on the theory of Hierarchically Separated Trees (HST's) introduced by Bartal [1996] in a seminal paper.

On the practical side, our experimental analysis with real and synthetic data shows that it is easy and inexpensive to compute multiple advice of the form described, allowing TAKEHEED to outperform state-of-the-art algorithms such as those in Meyerson [2001], Fotakis [2003], Anagnostopoulos et al. [2004]. Furthermore, we also show that we can make our algorithm robust to erroneous advice: when $\mathcal{S}$ is far from any optimal set of facilities for our input, $\text{OPT}(\mathcal{S})$ can be much larger than the actual offline optimum $\text{OPT}$. However, by combining TAKEHEED with known results of Mahdian et al. [2012] we can get a robust version of TAKEHEED called WARY, whose cost is

$$\mathcal{O}\left(\min\left\{\log(|\mathcal{S}|) \cdot \text{OPT}(\mathcal{S}), \frac{\log(n)}{\log\log(n)} \cdot \text{OPT}\right\}\right).$$

Note that the second term matches the best possible bound, in view of the lower bound of Fotakis [2003]. Our experiments confirm the good behavior in practice of this robust version. Indeed, it outperforms other state-of-the-art robust solutions.

One of the appealing features of TAKEHEED is that it leverages multiple advice in a simple way. The algorithm however could suffer from an "adversarial" attack of sorts. If very large bad sets of suggested facilities are given as part of the advice, the theoretical guarantee, which depends on the cardinality of the union of the advice sets, becomes uninteresting. To deal with this we design

BUCKETHEED, an algorithm that is resilient to the presence of such large, bad advice. If the best advice set is $\mathcal{S}_{i^*}$, the expected cost of the solution produced by this algorithm is

$$O\left(\log\log(n) \cdot (\log(k) + \log(|\mathcal{S}_{i^*}|)) \cdot \mathrm{OPT}(\mathcal{S}_{i^*})\right).$$

This result is mainly of theoretical interest. Indeed, one of the strengths of TAKEHEED is that in practice small advice sets are easy to obtain and the term $\log(|\mathcal{S}|)$ is going to be quite small, making the overall approach very actionable.

**Related Work.** Facility location and online algorithms have been extensively studied in the algorithm and machine learning literature. Here we give a high-level overview of the main results.

*Facility Location.* The offline metric (uncapacitated) facility location problem has been extensively studied, and various algorithms and heuristics have been formulated (see Korte and Vygen [2018]). The problem is known to be NP-hard, and it cannot be approximated within a factor of $1.463$ unless **NP** $\subseteq$ **DTIME**$(n^{\log\log(n)})$ [Guha and Khuller, 1999]. The best-known polynomial-time algorithm yields a $1.488$ approximation ratio [Li, 2013].

*Online Setting.* We cast our results in the standard setting of online competitive analysis [Borodin and El-Yaniv, 2005]. The online version of the metric facility location problem was introduced by Meyerson [2001], together with a randomized algorithm providing an expected $O(\log(n))$ approximation of the optimal solution. Subsequently, Fotakis [2003] provided a deterministic algorithm with a $O(\log(n)/\log\log(n))$ competitive ratio and proved that no randomized algorithm can obtain an asymptotically better competitive ratio against an oblivious adversary, even if the points lie on a line. Anagnostopoulos et al. [2004] also provided an online algorithm for facility location with restricted facilities: their algorithm works only in the euclidean plane, and our procedure combining the advice can be seen as a generalization of this result in a generic metric space. Fotakis [2005] formulated a different algorithm for the general online problem, achieving a $O(\log(n))-$approximation of the optimal cost with a primal-dual approach. Mahdian et al. [2012] showed how to combine two online algorithms for facility location into a single algorithm achieving a constant approximation of the costs of the best algorithm. In fact, the mixing procedure can be also be used to combine an algorithm with a suggested set of facilities, or two sets of suggested facilities. Applied inductively, it can also be used to mix multiple advice. In asymptotic terms, the resulting bound is incomparable to ours (see the Supplementary Material) and, in practice, our approach appears to be preferable, as documented in the experimental section. Other formulations of online clustering have been proposed: e.g., Mettu and Plaxton [2003] allow to chose the order by which processing the input, which is provided in advance; Guo et al. [2020] consider the online model with recourse, in which one is allowed to close some facilities and reassign the clients to new ones. Guo et al. [2020] also use HSTs for their clustering problem, but in a different way to ours.

*Online algorithms with learned advice.* There have been various previous attempts to improve online algorithms with external information. Common assumptions are related to the distribution of the input data. The most popular assumption is that data arrive according to the random arrival model, i.e. the input is a random permutation of a fixed unknown set of elements. This leads to major improvements in the competitive ratio for many problems [McGregor, 2014]. Even Meyerson's algorithm is shown to have a constant expected competitive ratio in this setting [Meyerson, 2001]. In many other scenarios, it is common to assume that input data are sampled from a fixed distribution, like for online matching [Mirrokni et al., 2012] or job scheduling [Mitzenmacher, 2020]. Recently, Lykouris and Vassilvitskii [2018] formulated the Online with Machine Learned Advice (OMLA) model, which is a theoretical framework for combining online algorithms with machine learning predictions. They also designed an algorithm for the caching problem in this model, which has been improved by Rohatgi [2020] and Wei [2020]. Various other problems have been tackled in this model with encouraging outcomes, like ski-rental [Purohit et al., 2018], the Secretaries problem [Dütting et al., 2020], and scheduling with restricted assignments [Lattanzi et al., 2020] (see Mitzenmacher and Vassilvitskii [2020] for a survey). In a slightly different setting, Medina and Vassilvitskii [2017] designed an algorithm for reserve price optimization in auctions. Learned predictions have been also used to improve classical data structures [Kraska et al., 2018, Mitzenmacher, 2018, Hsu et al., 2019]. A different model is the advice model, in which there is an oracle knowing the exact input and providing a small part of it to the online algorithm (see Boyar et al. [2017] for a survey). Note that this oracle is supposed to be perfect and completely reliable, while in our setting, the predictions can be extremely inaccurate, nevertheless the online algorithm needs to be robust to them. Another different setting is the one of online learning with experts [Choromanska and Monteleoni, 2012]:

here the advice, as the proposed clustering decisions, can change at every time step, while in our setting they cannot change over time.

*Online Facility Location with learned advice.* Very recent independent work in the same spirit as ours, made public on ArXiv after the present paper was submitted, are Fotakis et al. [2021] and Jiang et al. [2021].

## 2 Preliminaries

Let us begin by defining our problem: UNIFORM ONLINE FACILITY LOCATION. We are given a sequence of points $\mathcal{P}$, called *clients*, possibly with repetitions, from a metric space $X$ with distance $d$, and $f > 0$ called the uniform *facility cost*. The points are presented to us one after the other. As soon as a point $p$ arrives, we must select a point $c(p) \in X$, called an *opened facility*, and assign $p$ to it. $c(p)$ can be an already opened facility if it exists, or a brand new point. At the end of this procedure, every client in $p \in \mathcal{P}$ has a corresponding facility $c(p)$. Let $h$ be the number of opened facilities at the end. The cost of the assignment is defined as $h \cdot f + \sum_{p \in \mathcal{P}} d(p, c(p))$. If we consider facility location as a Lagrangian relaxation of $k-$median, then $f$ plays the role of Lagrangian multiplayer (the larger the $f$, the fewer the clusters). To identify an instance of the problem is sufficient to specify the pair $(\mathcal{P}, f)$ without explicit mention of the underlying metric space $(X, d)$, which is fixed and whose existence is clear from the context. We adopt this convention from now on. We denote as OPT$_\mathcal{P}$, or simply OPT when the context is clear, the cheapest possible facility assignment for $\mathcal{P}$. Sometimes the set of possible facilities will be restricted to be in a finite set $\mathcal{S}$. In this case, the optimum assignment will be denoted as OPT$(\mathcal{P}, \mathcal{S})$, or simply OPT$(\mathcal{S})$.

We make extensive use of the elegant theory of *Hierarchically Separated Trees* (HST's). The theory was introduced in a seminal paper by Bartal [1996]. In what follows we mainly use the definitions and results of Fakcharoenphol et al. [2004]. An *HST family* for a finite metric space $(X, d)$ consists of a family of rooted trees $\mathcal{T}$ and a probability distribution $\mathcal{D}$ over the trees of $\mathcal{T}$ satisfying the following properties. The leaves of every $T \in \mathcal{T}$ are the points of the metric space $X$. The edges of every tree in the family are weighted according to a simple scheme. First, the edges from a node to its children have the same weight. Second, if we follow any path from the root to a leaf the edge weights decrease exponentially. In this paper we only consider $2-$HSTs, so they always decrease by a factor of at least 2. Each tree $T$ induces a simple metric $d_T$ among the leaves: the distance between two leaves $u$ and $v$ is the sum of the edge weights along the unique path between them in $T$. Fakcharoenphol et al. [2004] show that, given any finite metric space $(X, d)$ it is possible to construct in polynomial-time a $2-$HST family $(\mathcal{T}, \mathcal{D})$ with this structure which moreover satisfies the following:

  i  for every $T \in \mathcal{T}$, and for all $u, v \in X$, $d(u, v) \leq d_T(u, v)$;

  ii  for all $u, v \in X$, $\mathbb{E}_{T \sim \mathcal{D}}[d_T(u, v)] \leq 8 \log(|X|) \cdot d(u, v)$.

All logarithms in this paper are to the base 2.

## 3 The Online Algorithm and Its Analysis

Recall our setting. We have an instance of the online facility location and multiple advice in the form of a family of sets, each suggesting a set of facilities to open. We would like to take advantage of these suggestions when they are good and ignore them when they are bad. The problem is that we do not know in advance which is which. The main technical tool toward the solution is the following.

**Theorem 3.1.** *Let $(\mathcal{P}, f)$ be an instance of uniform online facility location, let $\mathcal{S}_1, \ldots, \mathcal{S}_k$ be sets of facilities* (i.e. *finite sets of points from the underlying metric space), and let $\mathcal{S} := \bigcup_{i=1}^{k} \mathcal{S}_i$. Then, there is a randomized online algorithm that assigns facilities of $\mathcal{S}$ to points in $\mathcal{P}$ whose expected cost is*

$$O\left(\log(|\mathcal{S}|) \cdot \text{OPT}(\mathcal{P}, \mathcal{S})\right).$$

Intuitively, the sets $\mathcal{S}_1, \ldots, \mathcal{S}_k$ play the role of multiple machine-learned advice. We shall refer to each $\mathcal{S}_i$ as a *suggested set*. The first observation is that, OPT$(\mathcal{P}, \mathcal{S}) \leq \min_{i \in [k]} \text{COST}(\mathcal{P}, \mathcal{S}_i)$, where COST$(\mathcal{P}, X)$ denotes the cost incurred by using all and only the facilities of the set $X$. Thus, we can approximate well the best suggested set. In fact, no suggested set by itself needs to be particularly

good. As long as their union contains a good subset, a good solution can be found! For instance, every suggested set could contain a few good points and a lot of noise. Yet, the algorithm would be able to extract a good solution. Notice that, even if the advice comes from many sources, we combine it in a very simple way: we take the union $\mathcal{S}$ of the suggested sets. We believe this is quite an appealing feature of our approach. However, the $\log(|\mathcal{S}|)$ factor shows that we could suffer from the presence of a few large and bad advice sets. Even if this is not the case in practical settings, in the Supplementary Material we show how to deal with this problem.[1]

If the union $\mathcal{S}$ contains no good subset, OPT$(\mathcal{S})$ can be arbitrarily far from the true offline optimum OPT, so the previous solution can be bad. However, we can obviate the problem and attain robustness thanks to known results of Mahdian et al. [2012] and Meyerson [2001].

**Corollary 3.1.1.** *Let $\mathcal{P}$ and $\mathcal{S}$ be as in the statement of Theorem 3.1, and let $n = |\mathcal{P}|$. Then, there is an online randomized algorithm whose expected cost is at most,*

$$O\left(\min\left\{\log(|\mathcal{S}|)\cdot\text{OPT}(\mathcal{P},\mathcal{S}),\frac{\log(n)}{\log\log(n)}\cdot\text{OPT}\right\}\right).$$

Corollary 3.1.1 is a direct application of the results of Mahdian et al. [2012], and shows that our algorithm is comparable to the true offline optimum of the problem. Therefore, we focus on Theorem 3.1 from now on and we present the proof of Corollary 3.1.1 with the full description of the algorithm in the Supplementary Material.

Intuitively, we are going to implement the following reduction. The initial problem is to find good facilities, taken from the set $\mathcal{S}$, for an online sequence of points $\mathcal{P} := \{p_1,\ldots,p_n\}$. Instead, we will find facilities for the sequence of "nearby" points $Q := \{q_{p_1},\ldots,q_{p_n}\}$, where $q_p$ is the point of $\mathcal{S}$ closest to $p$. Recall that, since $\mathcal{S}$ is a metric space, there is an HST family $(\mathcal{T},\mathcal{D})$ for it. By construction, the leaves of every $T \in \mathcal{T}$ are the facilities in $\mathcal{S}$. To find good facilities for $Q$, we will pick a tree $T \in \mathcal{T}$ randomly according to $\mathcal{D}$ once and for all and use it to find good facilities for the entire sequence $Q$. When a request for $q_p$ arrives, we return a (hopefully) good leaf of $T$. In choosing the leaf, we work with the distance $d_T$ induced by the tree $T$ which is (in expectation) a good approximation of the distance $d$ in the metric space. And so, a good solution in the metric induced by $T$ is also going to be (in expectation) a good solution in the original metric space. The advantage of using $T$ relies on the strong structural properties of HSTs. Note that the sequence $Q$ may contain repeated points even when the input sequence $\mathcal{P}$ does not. Known online algorithms do not seem to be able to cope with this situation in a satisfactory way. Indeed, in Section 4 we prove a lower bound that rules out Meyerson's algorithm as well as other well-known algorithms, motivating our HST-based approach.

The core of the solution is algorithm PLUCK which finds a good facility for $p$'s proxy $q_p$. To find facilities for the whole input sequence, PLUCK is called repeatedly by procedure TAKEHEED. The input to TAKEHEED are the sequence of clients $\mathcal{P}$ from a metric space $(X,d)$, the set $\mathcal{S}$ of suggested facilities, and an HST family $(\mathcal{T},\mathcal{D})$ for the metric space $(\mathcal{S},d)$ induced by $\mathcal{S}$.

---

**Algorithm 1** Algorithm TAKEHEED

1: Pick $T$ at random from the HST family $(\mathcal{T},\mathcal{D})$.
2: When a point $p \in \mathcal{P}$ arrives, let $q_p \in \mathcal{S}$ be its closest suggested facility.
3: Call PLUCK with input $q_p, T$ and root$(T)$; assign the returned facility $\ell_p$ to $p$.

---

It remains to define PLUCK. Its input consists of a point $q_p \in Q \subseteq \mathcal{S}$, an HST tree $T$ as above, and a node $v \in T$, initially set to the root of $T$ (so, effectively, we can say that its input consists of $q_p$ and $T$). The output is a leaf $\ell \in T$. Starting from the root, PLUCK goes down the tree $T$ recursively in search of a good leaf for $q_p$. Once the leaf is found it is *opened* and returned. The tree $T$ is passed "by reference" so that: *once a leaf is opened, it remains open during all future invocations of* PLUCK. Initially, no leaf is open. Every node in the tree $T$ has a potential, initially set to zero. The potentials too are modified "by reference" and are preserved from one invocation of PLUCK to the next. In Algorithm 2 we give the pseudo-code for PLUCK. The main goal of PLUCK is to serve the newly arrived client with a facility within distance $f$ from $q_p$, with the additional guarantee that not too

---

[1]We develop another Algorithm, BUCKETHEED, which does not merge together all the advice sets, but assigns them to different "buckets" depending on their cardinality. In this way, it produces a solution with quality not depending on $|\mathcal{S}|$, but on the size of the best advice. See the Supplementary Material for more details.

many facilities are opened by successive executions of the algorithm. To achieve this goal PLUCK operates as follows. If there is no open facility within distance $f$ from $q_p$, PLUCK opens a new facility within distance $f/3$ from $q_p$. In so doing, it selects the facility that is able to serve the largest number of future clients (see subroutine SELECTHEAVIESTCHILD). Otherwise, PLUCK opens a new facility if and only if the potential of an internal node is higher than the cost of opening a facility. Intuitively, this guarantees that we do not open too many facilities because: in the former case we reduce the number of possible points without facilities within distance $f$ substantially; in the latter, we can charge the opening cost of the facility to the serving cost of the instance.

---

**Algorithm 2** Algorithm PLUCK$(T, v, q)$

---

**Input:** HST tree $T$, a node $v \in T$, a point $q \in \mathcal{S}$
**Output:** a leaf of $T$
 1: **if** $v$ is a leaf of $T$ **then**
 2:     open $v$ (if already opened, do nothing)
 3:     **return** $v$
 4: Let $T(v)$ be the sub-tree rooted at $v$
 5: **if** $T(v)$ has no opened leaf within dist. $f$ from $q$ **then**
 6:     $w = $ SELECTHEAVIESTCHILD$(T, v, q)$
 7:     **return** PLUCK$(T, w, q)$
 8: **else**
 9:     Let $x$ be the child of $v$ s.t. $q$ is a leaf of $T(x)$
10:     **if** $T(x)$ contains an opened leaf **then**
11:         **return** PLUCK$(T, x, q)$
12:     **else**
13:         Select the child $y$ of $v$ s.t. $T(y)$ contains the opened leaf $\ell$ closest to $q$ (ties broken arbitrarily)
14:         Increase $x$'s potential by $d_T(q, \ell)$
15:         **if** the new potential exceeds the cost $f$ of opening a facility **then**
16:             **return** PLUCK$(T, x, q)$
17:         **else return** $\ell$

---

PLUCK is well-defined and it always terminates (see the Supplementary Material for a formal proof).

**Notation 3.1.** *For every $q_p$ and tree $T$,* PLUCK *returns a leaf, denoted from now on as $\ell_p$, which is the facility associated to the point $p$ (recall that the leaves of $T$ are the set $\mathcal{S}$ of suggested facilities).*

Furthermore, note that $\ell_p$ is always at tree distance no greater than $f$ from $q_p$, *i.e.* $d_T(q_p, \ell_p) \leq f$. And, if $\ell_p$ is opened on input $q_p$ by the invocation of SELECTHEAVIESTCHILD, then $d_T(q_p, \ell_p) \leq f/3$.

---

**Algorithm 3** Algorithm SELECTHEAVIESTCHILD$(T, v, q)$

---

**Input:** an HST tree $T$, a node $v \in T$, and a point $q \in \mathcal{S}$
**Output:** a child of $v$ in $T$
 1: **return** $w$ such that:
        *(i)* $w$ is a child of $v$;
        *(ii)* $T(w)$ has a leaf at distance no greater than $f/3$ from $q$;
        *(iii)* the number of leaves of $T(w)$ is maximum (among those satisfying *(ii)*, breaking ties arbitrarily)

---

### 3.1 Algorithm Analysis

All the proofs have been moved to the Supplementary Material for lack of space. However, here we recreate the workflow of the proof of Theorem 3.1. To establish Theorem 3.1 we fix an optimal solution restricted to $\mathcal{S}$, $\mathcal{F}^* := \{c_1^*, \ldots, c_{k^*}^*\} \subseteq \mathcal{S}$, and compare it to the solution returned by TAKEHEED. The optimal solution naturally induces a clustering of the clients $\mathcal{C}^* := \{C_1^*, \ldots, C_{k^*}^*\}$ where $C_i^* := \{p \in \mathcal{P} : c_i^*$ is the facility in $\mathcal{F}^*$ closest to $p\}$.

**Notation 3.2.** *Let $c_p^*$ denote from now on the facility in the optimal solution closest to point $p$. And let $k^*$ denote the number of facilities opened by the optimal solution.*

In what follows, we establish a series of bounds that hold for every tree $T \in \mathcal{T}$ in the HST family. Once this is done, we invoke linearity of expectation to obtain the desired bound. Recall that $d$ is the distance in the original metric space while $d_T$ is the metric induced by a tree $T \in \mathcal{T}$. We start with two useful lemmas. The first says that we can bound the service cost of a point $p$ with that of its proxy $q_p$ in the tree metric $d_T$. The second relies on the properties of HSTs.

**Lemma 3.2.** *Let $T \in \mathcal{T}$. Then, $d(p, \ell_p) \leq d(p, c_p^*) + d_T(q_p, \ell_p)$.*

**Lemma 3.3.** $\mathbb{E}_{T \sim \mathcal{D}}[d_T(q_p, c_p^*)] \leq 16 \log(|\mathcal{S}|) d(p, c_p^*)$.

Hence, to bound the cost $d(p, \ell_p)$ for point $p$, we need to find a bridge between $d_T(q_p, \ell_p)$ and $d_T(q_p, c_p^*)$. We now partition the clients $\mathcal{P}$ into two groups, those that are "far" from the optimal centers and those that are "near": $\mathcal{P}_{\text{NEAR}}^T := \{p \in \mathcal{P} : d_T(q_p, c_p^*) \leq f/6\}$ and $\mathcal{P}_{\text{FAR}}^T := \mathcal{P} \setminus \mathcal{P}_{\text{NEAR}}^T$.

**Notation 3.3.** *Let $\ell_p$ be the opened facility returned by* PLUCK *on input $q_p$. If this is the first time the leaf was opened, we assign the entire facility cost $f$ to $p$. Otherwise, we assign zero. Let $f^T(p)$ denote the facility cost assigned to $p$ by this mechanism.*

Given that $T$ is fixed in our analysis and there is no ambiguity, we will drop the superscript $T$ for notational convenience. The cost incurred by far away points can be easily bounded through point-wise bounds, following by the definition of $\mathcal{P}_{\text{FAR}}$.

**Lemma 3.4.**

$$\sum_{p \in \mathcal{P}_{\text{FAR}}} \Big( d_T(q_p, \ell_p) + f(p) \Big) \quad \leq \quad 12 \sum_{p \in \mathcal{P}_{\text{FAR}}} d_T(q_p, c_p^*).$$

We can focus on $\mathcal{P}_{\text{NEAR}}$ from now on, for which the proofs get more involved. In Lemma 3.2 we bounded the service cost incurred by the algorithm for each point $p$ with its optimal cost and the distance between the proxy $q_p$ and the facility $\ell_p$. We now do the same for the facility cost.

**Lemma 3.5.** *Let $T \in \mathcal{T}$.*

$$\sum_{p \in \mathcal{P}_{\text{NEAR}}} f(p) \quad \leq \quad 3 \sum_{p \in \mathcal{P}} d_T(q_p, \ell_p) + f k^*$$

In view of Lemmas 3.2, 3.3 and 3.5, to close the circle we need to relate $d_T(q_p, \ell_p)$ to $d_T(q_p, c_p^*)$. Technically this is the most challenging step, achieved by the next lemma which requires careful use of the properties of PLUCK.

**Lemma 3.6.**

$$\sum_{p \in \mathcal{P}_{\text{NEAR}}} d_T(q_p, \ell_p) \leq 2 f k^* \log(|\mathcal{S}|) + 4 \sum_{p \in \mathcal{P}_{\text{NEAR}}} d_T(q_p, c_p^*).$$

By combining Lemmas $3.4 - 3.6$ a similar bound for the facility costs can be established.

**Lemma 3.7.** *Let $T \in \mathcal{T}$. Then,*

$$\sum_{p \in \mathcal{P}_{\text{NEAR}}} f(p) \leq \big( 6 \log(|\mathcal{S}|) + 1 \big) f k^* + 18 \sum_{p \in \mathcal{P}} d_T(q_p, c_p^*).$$

All the pieces can be assembled to prove Theorem 3.1 (see the Supplementary Material).

# 4  Lower Bounds

In this section we present two lower bounds whose proof is deferred to the Supplementary Material. The first shows the approximation guarantee of Theorem 3.1 to be almost optimal. The second shows that, in the worst case, Meyerson [2001]'s well-known algorithm computes a solution whose cost is $\Omega(|\mathcal{S}| \text{OPT}(\mathcal{S}))$, as opposed to the $O(\log(|\mathcal{S}|) \text{OPT}(\mathcal{S}))$ bound of Theorem 3.1. The same is also true for the algorithms from Fotakis [2003, 2005]. This provides strong motivation for our HST approach.

**Theorem 4.1.** *Let $f = 1$ be the uniform facility cost. For every $k \geq 1$, there exist a metric space, $k$ sets of facilities $\mathcal{S}_i, i \in [k]$, (with $|\mathcal{S}| = |\cup_{i \in [k]} \mathcal{S}_i| = k$), and a probability distribution over the input sequences such that, when an input sequence $\mathcal{P}$ is generated according to this distribution: (i) with probability 1, $\min_{i \in [k]} \text{COST}(\mathcal{P}, \mathcal{S}_i) = O(1)$ (i.e. one of the sets suggests a solution of cost $O(1)$) and yet, (ii) no online algorithm can provide a solution of expected cost smaller than $o\left(\frac{\log |\mathcal{S}|}{\log \log |\mathcal{S}|}\right)$.*

We now proceed by defining a general class of algorithms for online facility location.

Table 1: For the real datasets, an instance consists of the activities of one day. To obtain an input for the algorithm an instance is paired with facility cost.

| Dataset | # Instances | Median size | Size range |
|---------|-------------|-------------|------------|
| Mixtures | 750 | $5,000$ | $1,000$ - $10,000$ |
| Brightkite | 901 | $3,119$ | $5$ - $7,140$ |
| Gowalla | 388 | $9,354$ | $39$ - $25,425$ |
| Uber | 183 | $24,550$ | $10,202$ - $43,205$ |

**Definition 4.1.** *An algorithm for online facility location is said to be "Meyerson-like" if it only opens facilities at previously seen input points.*

Notice that the algorithms from Meyerson [2001], Fotakis [2003] are Meyerson-like according to Definition 4.1, while the algorithms by Anagnostopoulos et al. [2004], Fotakis [2005] are not.

**Theorem 4.2.** *Let $f = 1$ be the uniform facility cost. For every integer $k > 0$, there is a finite metric space $(\mathcal{S}, d)$ of cardinality $|\mathcal{S}| = k$ and input sequences such that: (i) The optimal solution has cost $O(1)$, but (ii) Any Meyerson-like algorithm computes solutions of cost $\Omega(|\mathcal{S}|) = \Omega(k)$; (iii) SNFL algorithm from Fotakis [2005] computes solutions of cost $\Omega(|\mathcal{S}|) = \Omega(k)$.*

Notice that $|\mathcal{S}| = k$ is not the size of the input, which could contain repeated points of $\mathcal{S}$, so these lower bounds have nothing to do with the logarithmic lower bound from Fotakis [2003]. Theorem 4.2 further validates our approach, since it rules out any Meyerson-like algorithm, which include many online algorithms for facility location, and SNFL algorithm from Fotakis [2005] too. On the other side, we manage to bypass this lower bound by considering as candidate facilities also points from the advice, and exploiting HSTs to identify the limit point in a subset of the facilities.

## 5   Experiments

We studied the performance of TAKEHEED with real-world and synthetic datasets by comparing it against the state of the art online algorithms of Meyerson [2001], Fotakis [2003], and Anagnostopoulos et al. [2004]. For all algorithms above we used our own implementation[2]. Note that the first algorithm is randomized, as it is TAKEHEED. To generate the trees of the HST families we implemented the algorithm of Fakcharoenphol et al. [2004].

As a baseline to compare against we use the 1.52-approximation algorithm for offline facility location by Mahdian et al. [2002], referred to as THEBASELINE in the sequel. For it, we used the Max Planck Institute implementation [Hoefer, 2002].

To test robustness, we use the algorithm of Mahdian et al. [2012]. This algorithm accepts in input two algorithms and mixes them in a way as to ensure that cost of its output is always of the same order of magnitude of the better of the two. The algorithm can also mix a set of points (*i.e.* advice) with an algorithm, or two sets of points.

For synthetic instances, we sampled points from a mixture of Gaussians (from 10 to 100) having centers in $[0, 1]^2$ and variance scaled by the number of mixtures. The metric is the euclidean distance. For real real-world datasets, we consider *Gowalla* and *Brightkite*, from the SNAP Dataset Collection [Leskovec and Krevl, 2014], and *Uber* [FiveThirtyEight, 2015]. Each of these contains latitude, longitude, and timestamp of some activity: check-in from the location-based social networks [Cho et al., 2011] for the first two (restricted to the US), and pickups in New York City for Uber. For each dataset, an input instance consists of all such activities that occurred in a given day, sorted by timestamp. The metric used is the great-circle distance in kilometers. Table 1 gives additional information on the datasets.

 For every input instance, all randomized algorithms were run 10 times. The plots report the average and variance of the cost of their solutions. All experiments ran on a desktop computer.

**Synthetic Instances.** The input sequence and the advice were generated as follows. First, a set of $N$ points was sampled from the mixture of Gaussians. The input sequence consisted of these points in random order. The advice was obtained as follows. THEBASELINE was given the random set in

---

[2]https://github.com/matteojug/Online-Facility-Location-with-Multiple-Advice

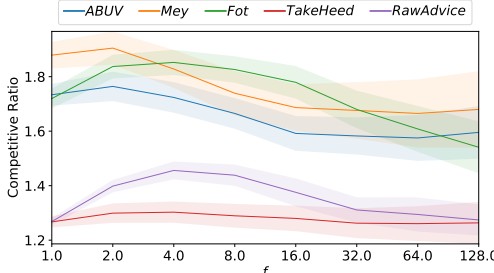 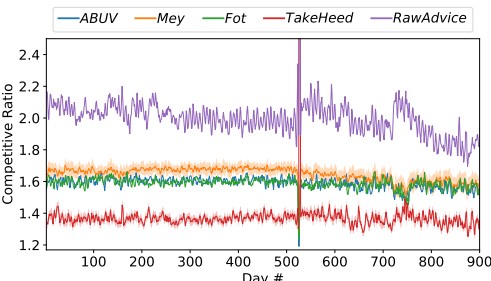

Figure 1: Average estimated competitive ratios on Gaussian mixtures (5000 points from 50 Gaussians) with noisy advice. The shaded area represents one standard deviation.

Figure 2: Average estimated competitive ratios on Brightkite with $f = 1000$. The shaded area represents one standard deviation.

input and produced a set of $M$ facilities in output. Then, $M$ additional random points were added. Finally, the $2M$ points were randomly partitioned into 5 sets of equal size. In the experiments, for every mixture, 50 random instances were generated.

**Real-World Instances.** For real-world datasets, we divided up every day into four consecutive time windows of 6 hours each, starting from midnight. An input sequence consisted of all the points (activities) of a given day. The multiple advice was obtained by running THEBASELINE on each time window of the two previous days, for a total of 8 sets of suggested facilities. In this way, we were able to obtain multiple advice containing both bad and good suggestions. For every input sequence, we tried different values of $f$, the facility cost. Changing $f$ did not affect the outcome in terms of performance comparison. The results we show exemplify the general picture. Likewise, we ran all the randomized algorithms using different seeds. Again, this had no discernible consequence on the performance.

**Raw advice.** The previous discussion describes how multiple advice was generated. Algorithm RAWADVICE provides a useful baseline to quantify its quality. It operates as follows. Given the union of the facility sets comprising the advice, each client in the input sequence is assigned to the closest facility, which is opened. The cost of this solution gives a useful quantitative indication of how good the multiple advice is. In principle, the algorithm of Mahdian et al. [2012] gives another way to mix the sets. By generalizing the mixing process, it is possible to work with multiple algorithms, switching to the best-performing one for each arriving client. In practice, however, just taking the union gives the best result. This is true for both the normal and robust versions of the algorithm. So we focus only on the approach merging all the suggestions in what follows.

**Preprocessing.** The performance of TAKEHEED hinges upon the tree sampled from the HST family (recall that we used the algorithm from Fakcharoenphol et al. [2004]). Recall that the points of the multiple advice induce a metric space for which an HST family exists. To improve performance, we sampled several trees (10 and 100, respectively, for real and synthetic datasets) and picked the one that minimized distortion[3]. The distortion was computed naïvely, as the maximum over all the pairs of points, with no attempt at optimization. Even so, the preprocessing time was negligible. We do not report exact figures because here we are only concerned with the cost of the solutions.

**Outcome.** In the sequel we use the term *competitive ratio* in the following sense: the competitive ratio of algorithm X is the ratio between the cost of the solution given by X and that of THEBASELINE. This gives a uniform measure of comparison. For Fotakis and ABUV algorithms, which are parametrized, we tried different values for the parameters and report here only the best ones.

*Synthetic Instances.* In Figure 1 we report the results on synthetic instances. Note that the number of opened facilities for large values of $f$ – the opening cost – becomes small and the difference in behavior between TAKEHEED and RAWADVICE decreases. In presence of noisy advice, TAKEHEED consistently outperforms both the other online algorithms and the naive use of the advice. Similar results are obtained for different numbers of Gaussian mixtures.

---

[3]the ratio between the metric space distance and the tree distance, for the worst pair of points.

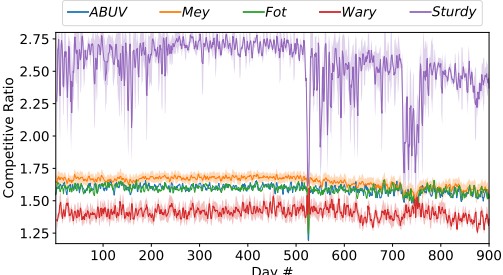
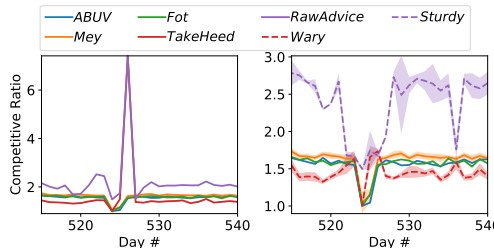

Figure 3: Average estimated competitive ratios of the robust variants and the baselines on Brightkite with $f = 1000$. The shaded area represents one standard deviation.

Figure 4: Detail of 2 and 3 around some extreme instances (not smooth). The robust versions manage to recover from bad advice.

*Real-World Instances.* In Figure 2 we compare TAKEHEED with the competitors for the Brightkite dataset. Similar results were obtained with all datasets (see the Supplementary Material). The facility cost was determined in such a way that THEBASELINE opened a number of facilities between 1%–10% of the number of input points. TAKEHEED always outperforms all other algorithms, except for one outlier instance, to which we return (see the spike occurring around $day = 525$ of the plot). This good behavior was consistent across all datasets, illustrating the interesting fact that high-quality multiple advice might be cheaply and consistently produced.

**Robustness.** As mentioned, robust online algorithms can be obtained by using the mixing algorithm of Mahdian et al. [2012]. Algorithm WARY is obtained by mixing our TAKEHEED with Meyerson's algorithm. While algorithm STURDY is obtained by simply mixing Meyerson's algorithm with the advice set. The mixing algorithm has a parameter $\gamma$ with which one can give more weight to one of the two components to be mixed. We show the outcome for $\gamma = 1.75$ which gave best results ($\gamma = 2$ treats them equally, while $\gamma = 1.75$ gives less weight to Meyerson's algorithm). An in-depth analysis of the role of this parameter is given in the Supplementary Material.
Figure 3 reports the outcome for the same Brightkite dataset, while Figure 4 zeroes in on the spike of Figure 2. The latter illustrates how the robustness plays out when the advice is bad. While TAKEHEED suffers from receiving a bad advice, its robust counterpart WARY does as well as the best online algorithms using no advice, as predicted by the theory. Different values of the facility cost $f$ produced the same outcome.

## 6 Conclusion

We introduced TAKEHEED, WARY, and BUCKETHEED, three online algorithms for online facility location that take advantage in different ways of multiple advice – *i.e.* advice coming from disparate sources possibly containing noise and misleading information. The algorithms exhibit good theoretical guarantees as well as good practical performance. Our approach also illustrates that high-quality multiple advice might be easy and inexpensive to obtain in practical situations, giving rise to algorithms that can outperform online algorithms that heed no advice. In those seemingly rare occasions in which good advice is not possible to obtain, WARY, the robust version of TAKEHEED, has been shown to be able to disregard the poor advice and perform as well as the best online algorithms. We consider our results as a case study of a more general philosophy that appears to be a promising approach to tackle a large variety of interesting problems.

## Acknowledgements

This work was supported in part by a Google Focused Research Award, by the PRIN project 2017K7XPAN, by BiCi — Bertinoro international Center for informatics, and by the MIUR under grant "Dipartimenti di eccellenza 2018-2022" of the Department of Computer Science of Sapienza University.

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
