# Online Facility Location with Multiple Advice
## Supplementary Material

## A  Properties of HSTs

We present a few useful facts on HSTs for later use. First, for the HST scheme from Fakcharoenphol et al. [2004], the following fact stands.

**Fact A.1.** *Let $T \in \mathcal{T}$. Then, the edges from any internal node of the tree to its children have all equal lengths.*

From this peculiar property of the HSTs generated by the Algorithm by Fakcharoenphol et al. [2004], two useful lemmas follow.

**Lemma A.2.** *Let $T \in \mathcal{T}$, let $x$ and $y$ be any two leaves of $T$, and let $v$ be their lowest common ancestor. Then, $d_T(x,v) \geq d_T(v,y)/2$ and $d_T(x,v) \geq d_T(x,y)/3$.*

*Proof.* By the triangle inequality, $d_T(x,y) \leq d_T(x,v) + d_T(v,y)$. Now, by Fact A.1, all the edges from $v$ to its children have all equal length, name it $d_v$, so $d_T(x,v) \geq d_v$. Moreover, by definition of $2-$HST, the edge lengths decrease geometrically by a factor $\geq 2$ on any root to leaf path. Therefore, we have that $d_T(v,y) \leq d_v + \sum_{i=1}^{\infty} d_v/2^i = 2d_v$, so $d_T(v,y) \leq 2d_T(x,v)$. By substituting this above, we also get that $d_T(x,y) \leq 3d_T(x,v)$, as desired. $\qquad\square$

**Lemma A.3.** *Let $T \in \mathcal{T}$, let $v_2$ be any node, $v_1$ one of its descendants, and $v_0$ a descendant of $v_1$. Then, $d_T(v_0,v_1) \leq d_T(v_0,v_2)/2$.*

*Proof.* By Fact A.1, all the edges from $v_2$ to its children have all equal length, name it $x$, so $d_T(v_1,v_2) \geq x$. Moreover, by definition of $2-$HST, the edge lengths decrease geometrically by a factor $\geq 2$ on any root to leaf path. Therefore, we have that $d_T(v_0,v_1) \leq \sum_{i=1}^{\infty} x/2^i = x$, so $d_T(v_0,v_2) = d_T(v_0,v_1) + d_T(v_1,v_2) \geq d_T(v_0,v_1) + x \geq 2d_T(v_0,v_1)$, as desired. $\qquad\square$

**Lemma A.4.** *Let $x,y,z$ be leaves of $T$. Let $r_1$ be the lowest common ancestor of $x,y$ and let $r_2$ be the lowest common ancestor of $x,z$ in $T$. Assume that $r_1$ is a descendant of $r_2$. Then, $d_T(x,y) \leq d_T(x,z)$.*

*Proof.* By Lemma A.2 we have that $d_T(x,y) \leq 3d_T(x,r_1)$. Moreover, by Lemma A.3, we have that $d_T(x,r_1) \leq d_T(x,r_2)/2$. Thus, by putting them together, we get that $d_T(x,y) \leq \frac{3}{2}d_T(x,r_2)$. It would suffices to show that $d_T(x,z) \geq \frac{3}{2}d_T(x,r_2)$. Now, $d_T(x,z) = d_T(x,r_2) + d_T(r_2,z)$ and, by Lemma A.2, it holds $d_T(r_2,z) \geq d_T(x,r_2)/2$, implying that $d_T(x,z) \geq \frac{3}{2}d_T(x,r_2)$ indeed. $\qquad\square$

We also report the proofs of the results in the main paper that link a metric space with its HST approximated embedding.

PROOF OF LEMMA 3.2.  By the triangle inequality and by definition of $q_p$, we have that $d(p,\ell_p) \leq d(p,q_p)+d(q_p,\ell_p) \leq d(p,c_p^*)+d(q_p,\ell_p)$. Moreover, by the properties of $T$ we have that $d(q_p,\ell_p) \leq d_T(q_p,\ell_p)$. And so, $d(p,\ell_p) \leq d(p,c_p^*) + d_T(q_p,\ell_p)$. $\qquad\square$

PROOF OF LEMMA 3.3.  $\mathbb{E}_{T \sim \mathcal{D}}\big[d_T(q_p,c_p^*)\big] \leq 8\log(|\mathcal{S}|)d(q_p,c_p^*)$ since $(\mathcal{T},\mathcal{D})$ is an HST family. By the triangle inequality and by definition of $q_p$, $d(q_p,c_p^*) \leq d(q_p,p) + d(p,c_p^*) \leq 2d(p,c_p^*)$. $\qquad\square$

# B   Properties of Algorithm TAKEHEED

We start by showing that Algorithm PLUCK is well-defined.

**Lemma B.1.** *The invocation of Algorithm* PLUCK *to a leaf $q$ of a tree $T$ always returns a leaf $\ell$ of $T$ where a facility has been opened.*

*Proof.* Algorithm PLUCK and all its recursive calls receive as input:

- a tree $T$;
- a node $v$ of $T$, which is the root of $T$ at the first invocation;
- a leaf $q$ of $T$.

Consider the first invocation of algorithm PLUCK. If the tree $T$ consists only of a single node $v$, which is also a leaf, we execute lines 1–3 and return a leaf $v$ with opened facility. Otherwise, $v$ is the root of $T$ and is an internal node of $T$.

Now, assume $T(v)$ has no opened leaf within distance $f$ from $q$. In this case, we execute lines $6-7$: we get a child $w$ of $v$ from procedure SELECTHEAVIESTCHILD, and we recur on $w$. Notice that $T(w) \subseteq T(v)$ cannot have an opened leaf within distance $f$ from $q$, so we will do the same for every recursive call from now on until we get to a leaf $\ell$. At that point, we execute lines 1–3 and return $\ell$ with opened facility.

Finally, assume $T(v)$ has an opened leaf within distance $f$ from $q$. Now, consider the set $\mathcal{L}$ of opened leaves within distance $f$ from $q$ (this set is not empty by our last assumption), and let $r$ be the internal node of $T$ such that: (i) it has both $q$ and a leaf from $\mathcal{L}$ among its descendants; (ii) among the ones satisfying (i), it has minimum edge-distance from $q$. By definition of PLUCK, we will keep executing lines 9–11 until we reach $r$, because every subtree of $T$ containing $r$ has both $q$ and a leaf within distance $f$ from $q$ as descendants. When we get to $r$, we execute lines 9–10, but we skip line 11 because $x$, as defined in line 9, does not contain an element of $\mathcal{L}$. Therefore, we execute lines 12–14: we take a specific $\ell \in \mathcal{L} \cap T(r)$ and we increase the potential stored in $r$ by $d_T(q, \ell)$. Now, we check the condition at line 15: if the potential stored in $r$ does *not* exceed $f$, we execute line 17 and just return the opened leaf $\ell$, and we are done. Otherwise, we execute line 16 and we recur on $T(x)$. Now, by definition of $r$, $T(x)$ does not contain any opened leaf within distance $f$ from $q$. So, we get back to the previous case: we keep executing recursively lines 6–7 until we get to a leaf, for which we execute lines 1–3 to return an opened leaf. All the possible cases have been covered. □

From the analysis of the workflow done in the previous proof, we can derive the following corollary.

**Corollary B.1.1.** *The invocation of Algorithm* PLUCK *to a leaf $q$ of a tree $T$ always affects the potential of at most one internal node.*

*Proof.* By the proof of Lemma B.1, we could only increase the potential of the aforementioned node $r$, which is the lowest common ancestor of $q$ and the leaves of the set $\mathcal{L}$ of opened facilities within distance $f$ from $q$. □

# C   Proofs in the sampled HST

Here we present the missing proofs from the analysis of our algorithm for points in $\mathcal{P}_{\text{FAR}}$. They follow by definition of $\mathcal{P}_{\text{FAR}}$.

**Lemma C.1.** *Let $T \in \mathcal{T}$ and let $p \in \mathcal{P}_{\text{FAR}}$. Then, (i) $f(p) \leq 6d_T(q_p, c_p^*)$, and (ii) $d_T(q_p, \ell_p) \leq 6d_T(q_p, c_p^*)$.*

*Proof.* By definition of $\mathcal{P}_{\text{FAR}}$, $f(p) \leq f \leq 6d_T(q_p, c_p^*)$ and the first claim follows. For the second, recall that PLUCK always returns a leaf within tree distance at most $f$. Hence, $d_T(q_p, \ell_p) \leq f \leq 6d_T(q_p, c_p^*)$. □

The previous Lemma can be used to prove Lemma 3.4.

PROOF OF LEMMA 3.4. By Lemma C.1, for every $p \in \mathcal{P}_{\text{FAR}}$, $d_T(q_p, \ell_p) + f(p) \leq 12\, d_T(q_p, c_p^*)$. The claim follows. $\qquad\square$

We also present the missing proofs from the analysis of our algorithm for points in $\mathcal{P}_{\text{NEAR}}$.

First, we define the restricted optimal clusters $\mathcal{R}^* = \{R_1^*, \ldots, R_{k^*}^*\}$, where

$$R_i^* := C_i^* \cap \mathcal{P}_{\text{NEAR}} = \{p \in C_i^* \mid d_T(q_p, c_i^*) < f/6\}.$$

We show that, for most points in $\mathcal{P}_{\text{NEAR}}$, we open a new facility when the potential of an internal node exceeds $f$. In this case, the cost of opening the new facility is amortized across the set of points whose potential was accumulated in that internal node.

PROOF OF LEMMA 3.5. Fix a generic $1 \leq i \leq k^*$ and focus on the restricted cluster $R_i^*$. We want to show that

$$\sum_{p \in R_i^*} f(p) \leq f + 3 \cdot \sum_{p \in X_i^*} d_T(q_p, \ell_p), \tag{1}$$

where $X_i^* \subseteq P$ is a set depending on $R_i^*$. Moreover, for two different restricted clusters $R_i^*, R_j^*$, the corresponding sets are disjoint: $X_i^* \cap X_j^* = \emptyset$. Therefore, once Equation (1) is established, the claim follows by summing over all clusters. If points in $R_i^*$ give rise to zero or one opened facility the claim follows. Assume therefore that they bring about two or more openings. Let $p_0, \ldots, p_k$ be the points in $R_i^*$ such that each one of them gives rise to a new opened leaf, denoted as $\ell_0, \ldots, \ell_k$. And let $q_0, \ldots, q_k$ be a shorthand notation for their proxies $q_{p_0}, \ldots, q_{p_k}$. Recall that for every $p \in R_i^*$ there is an invocation of PLUCK with its proxy $q_p$ in input.

First, observe that $d_T(q_0, \ell_0) \leq f/3$. This is because when PLUCK opens a new leaf that is always within distance $f/3$ from its client.

By the triangle inequality and the definition of $R_i^*$ we have that, for all $j \in [k]$,

$$d_T(q_j, \ell_0) \leq d_T(q_j, c_i^*) + d_T(c_i^*, q_0) + d_T(q_0, \ell_0) \leq \frac{f}{6} + \frac{f}{6} + \frac{f}{3} = \frac{2}{3} \cdot f. \tag{2}$$

Focus now on $p_1$. We want to upper bound its contribution to the facility cost (*i.e.*, $f(p_1)$) in terms of the service costs of some points processed before $p_1$ for which no new facility is opened. By definition of PLUCK, when called on $q_1$, the algorithm executes lines 13 and 14, and it does so exactly once by Corollary B.1.1. Let $x_1$ be the node defined in line 9 whose potential is incremented in line 14. Since $p_1$ caused the opening of a leaf, $x_1$'s potential is greater than $f$ after $q_1$ is processed (line 15). By Equation (2), $d_T(q_1, \ell_0) \leq 2/3 \cdot f$, and this is the increment of potential due to $q_1$, and so the potential of $x_1$ must be at least $f/3$ by the time $q_1$ is processed. Observe that the potential of any node is increased at most once for every call of PLUCK by Corollary B.1.1. So the increase can be ascribed to a specific set of points processed before $p_1$. For a point $p$ such an increment is, by definition of PLUCK, $d_T(q_p, \ell_p)$ (line 14). Denoting as $P_1^i$ the points processed before $p_1$ that affected $x_1$'s potential, we have,

$$\frac{f}{3} \leq \sum_{p \in P_1^i} d_T(q_p, \ell_p)$$

And therefore, $f(p_1) = f \leq 3 \cdot \sum_{p \in P_1^i} d_T(q_p, \ell_p)$. We can repeat the same argument for $p_2$, and so on so forth. We will end up defining the sets $P_j^i$, consisting of all the points processed before $p_j$ that affected $x_j$'s potential, where $x_j$ is defined analogously to $x_1$. Those sets $\{P_j^i\}_j$ are all disjoint, because they are the set of vertices affecting the potential of distinct nodes, and each client affects the potential of *at most one* node of the HST by Corollary B.1.1. Thus,

$$\sum_{p \in R_i^*} f(p) = f(p_0) + \sum_{j=1}^{k} f(p_i) \leq f + 3 \cdot \sum_{j=1}^{k} \sum_{p \in P_j^i} d_T(q_p, \ell_p) \leq f + 3 \cdot \sum_{p \in X_i^*} d_T(q_p, \ell_p),$$

where $X_i^* := \bigcup_{j=1}^{k} P_j^i \subseteq P$. This establishes Equation (1); the sets $\{X_i^*\}_i$ defined for different restricted clusters are disjoint, because so are the sets $\{P_j^i\}_{i,j}$. The claim follows. $\qquad\square$

Before proving the bound for $\sum_{p \in \mathcal{P}_{\text{NEAR}}} d_T(q_p, c_p^*)$, we need an auxiliary lemma.

**Lemma C.2.** *For every client $p$, let $\mathcal{F}_p$ be the set of opened facilities after running procedure* PLUCK *with input $q_p$. Then $\forall\, q \in \mathcal{F}_p$ we have $d_T(q_p, \ell_p) \leq d_T(q_p, q)$.*

*Proof.* The lemma is trivial if $\ell_p$ is the only opened facility within distance $f$ from $q_p$. Moreover, if $q_p$ does not open any new facility, the lemma follows because, by Lemma A.4 and by definition of PLUCK, the returned facility $\ell_p$ has minimum distance from $q_p$ among the opened facilities in $\mathcal{F}_p$. Assume this is not the case, so $\ell_p$, returned by procedure PLUCK, has just been opened as facility. Now, let $r$ be the lowest common ancestor of $\ell_p, q_p$. By definition of the procedure PLUCK, the only opened facility which has distance $\leq f$ from $q_p$ and is also a descendant of $r$ is $\ell_p$. Moreover, $\ell_p$ is closer to $q_p$ than all the other facilities in $\mathcal{F}_p \setminus T(r)$ by Lemma A.4, so we are done. $\qquad\square$

We now provide the fundamental bound for $\sum_{p \in \mathcal{P}_{\text{NEAR}}} d_T(q_p, c_p^*)$. We do this on each restricted cluster.

The general idea behind the following Lemma is inspired by the approximation ratio proof for Meyerson's Algorithm from Fotakis [2011]. In that proof, for each cluster, one seeks to bound the service cost by analyzing the input in phases, where a new phase begins when a new facility is opened at a point that is significantly closer to the center of the optimal cluster. Moreover, the cost of the points seen in each phase can be easily bounded. In their argument, it is essential to notice that the number of phases is logarithmic in the input size. In our proof, we follow a similar strategy, but we use restricted clusters instead of the whole clustering. Moreover, procedure SELECTHEAVIESTCHILD makes sure that the number of phases is logarithmic in $|\mathcal{S}|$ because it roughly halves the HST every time we open a facility at a leaf that is significantly closer to the center of the restricted cluster.

**Lemma C.3.** *For every $T \in \mathcal{T}$ and every restricted optimal cluster $R_i^*$ we have,*

$$\sum_{p \in R_i^*} d_T(q_p, \ell_p) \leq 2 \log(|\mathcal{S}|) f + 4 \sum_{p \in R_i^*} d_T(q_p, c_i^*). \tag{3}$$

*Proof.* Let $r$ be the lowest common ancestor in $T$ of all the leaves in $R_i^*$; let $T(r)$ be the sub-HST of $T$ rooted at $r$ and containing all its descendants, and let $h$ be its height. Clearly, all the points in $R_i^*$ are contained in this tree, but there could be some other leaves. For the sake of the cost, we will restrict our attention only to the leaves in $R_i^*$. We will consider $h + 2$ phases: $h, h - 1, \ldots, 0, -1$. We say that a point $p \in R_i^*$ belongs to phase $j$, i.e. $p \in \text{phase}_j$, if, when $p$ arrives in input, the largest sub-HST of $T(r)$ containing $c_i^*$ and no opened facilities has height $j$. We name this tree $\text{tree}_j$. In the proof, we keep track of these trees and show that when we go from one phase to the next their size shrinks by at least a half, which implies that there can be at most a logarithmic number of phases. We then focus on each phase to bound the service cost. We start with phase $h$ when no facilities are opened in $T(r)$; all the points seen after the possible opening of a facility at $c_i^*$ belong to phase $-1$. We say that a phase is *empty* if no client in $R_i^*$ is seen during that phase, i.e. $\text{phase}_j = \emptyset$. We proceed by proving simpler facts, which will imply the lemma altogether.

**Fact 0.** *Empty phases have service cost contribution 0 to $\sum_{p \in R_i^*} d_T(q_p, \ell_p)$.*
No client in $R_i^*$ is seen in an empty phase, so there is no contribution to the sum.

**Fact 1.** *For each pair of leaves $q_1, q_2 \in T(r)$, we have that: $d_T(q_1, c_i^*) \leq f/3$, $d_T(q_1, q_2) \leq 2/3 \cdot f < f$.*
Recall that $r$ is the root of $T(r)$. First, there exists a leaf $q' \in T(r)$ s.t. $d_T(c_i^*, q') \leq f/6$ and whose lowest common ancestor with $c_i^*$ is $r$. This is true because $r$ is the lowest internal node of $T$ containing all the points in $R_i^*$ as descendants. Now, since $r$ is in the path between $c_i^*$ and $q'$, we have that $d_T(c_i^*, q') = d_T(c_i^*, r) + d_T(r, q')$. By Lemma A.2, we have that $d_T(r, q') \geq d_T(c_i^*, r)/2$, so $f/6 \geq d_T(c_i^*, q') = d_T(c_i^*, r) + d_T(r, q') \geq \frac{3}{2} \cdot d_T(c_i^*, r)$, implying that $d_T(c_i^*, r) \leq f/9$. Now take a generic leaf of the tree, name it $q_1$: by Lemma A.2 we have that $d_T(q_1, c_i^*) \leq 3 \cdot d_T(c_i^*, r) \leq f/3$. Therefore, every leaf of the tree is within HST distance $\leq f/3$ from $c_i^*$. We can conclude by the triangle inequality that $d_T(q_1, q_2) \leq d_T(q_1, c_i^*) + d_T(c_i^*, q_2) \leq f/3 + f/3 = 2/3 \cdot f < f\ \forall\, q_1, q_2$ leaves of $T(r)$.

**Fact 2.** *Consider a generic client $p$ seen in a phase $0 \leq j < h$. If such a point opens a facility $\ell_p$ in this tree, then $q_p$ belongs to the largest sub-HST of $T(r)$ that contains $\ell_p$, but did not contain any opened facility before seeing $p$.*

Consider the smallest sub-HST of $T(r)$ including $\ell_p$ and another opened facility in $T(r)$: let $\bar{q}$ be such opened facility and let $r'$ be the root of this tree. When opening $\ell_p$, the procedure PLUCK visited $r'$ and decided to go in direction of $\ell_p$. Since, by Fact 1, $q_p$ is within distance $\leq f$ from $\bar{q}$, this means that the procedure was called on a point belonging to the same sub-HST of $T(r)$ containing $\ell_p$, which is indeed the largest sub-HST of $T(r)$ (containing $\ell_p$) that did not contain any opened facility before seeing $p$.

**Fact 3.** *Let $0 \leq j' < j \leq h$ and suppose we switch from phase $j$ to phase $j'$. Then, $|tree_{j'}| < |tree_j|/2$.*
Let $\ell$ be the leaf opened during phase $j$ that brings to phase $j'$. Consider the smallest sub-tree containing $c_i^*$ and $\ell$, and let $r'$ denote its root. And let $x$ and $y$ be the children of $r'$ such that $c_i^* \in T(x)$ and $\ell \in T(y)$. Since PLUCK goes down the tree and opens a new facility only when it reaches a leaf, PLUCK must have visited $r'$. When this happened, $T(r')$ had no opened leaves whatsoever since $T(r') \subseteq tree_j$ and $tree_j$ has no opened leaves by definition. And so line 6 was executed. Since PLUCK went all the way down to open $\ell$, SELECTHEAVIESTCHILD returned $y$ and discarded $x$. The latter follows from Fact 1, which implies that $x$ satisfies conditions (i) and (ii) in line 1 of SELECTHEAVIESTCHILD because $c_i^* \in T(x)$. Hence, $|T(x)| \leq |T(y)|$ and the claim follows, for $tree_{j'} \subsetneq T(x)$.

**Fact 4.** *There are at most $\log(|\mathcal{S}|)$ non-empty phases $j \geq 0$.*
It follows from Fact 3 which says: if $j' < j < h$, then $|tree_{j'}| < |tree_j|/2$.

**Fact 5.** *For a phase $j \geq 0$, it holds $\sum_{p \in phase_j} d_T(q_p, \ell_p) \leq 2 \cdot f + 4 \cdot \sum_{p \in phase_j} d_T(q_p, c_i^*)$.*
Consider a generic point $p$ in phase $j > 0$. Let $r_j$ be the root of $tree_j$. There can be two options for $p \in phase_j$:

- $q_p \notin tree_j$. This can only happen if $j < h$. By definition, we know that the father node $r_j'$ of $r_j$ belongs to $T(r)$ and has a child sub-tree containing an opened facility $\bar{q}$. First, since $\ell_p$ returned by Algorithm PLUCK is always the closest opened facility to $q_p$ by Lemma C.2, it holds $d_T(q_p, \ell_p) \leq d_T(q_p, \bar{q})$. Second, $d_T(q_p, \bar{q}) \leq d_T(q_p, c_i^*) + d_T(c_i^*, \bar{q})$ by the triangle inequality. Third, we have that $d_T(c_i^*, \bar{q}) \leq 3 \cdot d_T(q_p, c_i^*)$. This is true because $d_T(q_p, c_i^*) \geq d_T(c_i^*, r_j')$ since $r_j'$ belongs to the path from $q_p$ to $c_i^*$, and $d_T(c_i^*, r_j') \geq d_T(c_i^*, \bar{q})/3$ by Lemma A.2. Thus, $d_T(q_p, \ell_p) \leq d_T(q_p, \bar{q}) \leq 4 \cdot d_T(q_p, c_i^*)$;

- $q_p \in tree_j$. For all those points, except the last one, we can pay at most $f$ in terms of HST distances from the closest opened facility before opening a new facility in this tree, because this sum of distances is *all* cumulated as potential in the root of the tree. This is true because $r_j$ has a sibling (child of the same parent node) with an opened facility as a descendant and within distance $\leq f$ from $q_p$ by Fact 2, so the procedure PLUCK necessarily accumulates the corresponding potential in $r_j$. There can be a final point that induces the opening of a new facility corresponding to a change in the phase: this point, name it $p_{\text{LAST}}$, satisfies $d_T(q_{p_{\text{LAST}}}, \ell_{p_{\text{LAST}}}) \leq f$ as any other point. Therefore $\sum_{p \in tree_j} d_T(q_p, \ell_p) \leq f + f = 2 \cdot f$.

**Fact 6.** *For each point $p \in phase_{-1}$ it holds $d_T(q_p, \ell_p) \leq d_T(q_p, c_i^*)$.*
This follows because $c_i^*$ has been opened at this point so, since $\ell_p$ is the closest opened facility to $q_p$ by Lemma C.2, it holds $d_T(q_p, \ell_p) \leq d_T(q_p, c_i^*)$.

**Fact 7.** *It holds $\sum_{p \in R_i^*} d_T(q_p, \ell_p) \leq 2 \log(|\mathcal{S}|) \cdot f + 4 \cdot \sum_{p \in R_i^*} d_T(q_p, c_i^*)$.*
This follows by putting all the previous facts together:

$$\sum_{p \in R_i^*} d_T(q_p, \ell_p) = \sum_{\substack{j: \\ phase_j \neq \emptyset}} \sum_{p \in phase_j} d_T(q_p, \ell_p) \leq \sum_{\substack{j: \\ phase_j \neq \emptyset}} \sum_{p \in phase_j} \left( 2 \cdot f + 4 \cdot \sum_{p \in phase_j} d_T(q_p, c_i^*) \right).$$

where the inequality follows by Fact 5 and Fact 6. Since the union of all non-empty phases yields $R_i^*$ and, by Fact 4, there are $\leq \log(|\mathcal{S}|)$ non-empty phases $\geq 0$, and the bound for phase $-1$ comes from Fact 6, we finally get that

$$\sum_{p \in R_i^*} d_T(q_p, \ell_p) \leq 2 \log(|\mathcal{S}|) \cdot f + 4 \cdot \sum_{p \in R_i^*} d_T(q_p, c_i^*). \qquad \square$$

The proof of Lemma C.3 shows why procedure SELECTHEAVIESTCHILD always moves to the subtree with the largest number of leaves: this allows to have only $O(\log(|\mathcal{S}|))$ non-empty phases. Thus, even if the HST has depth $\Omega(|\mathcal{S}|)$, which is actually the case for some metrics, we are always able to navigate it with $O(\log(|\mathcal{S}|))$ steps.

Thanks to Lemma C.3, by summing over all the restricted clusters, we finally get Lemma 3.6.

PROOF OF LEMMA 3.6. By Lemma C.3 it holds

$$\sum_{p \in \mathcal{P}_{\text{NEAR}}} d_T(q_p, \ell_p) = \sum_{i=1}^{k^*} \sum_{p \in R_i^*} d_T(q_p, \ell_p) \leq \sum_{i=1}^{k^*} \left( 2\log(|\mathcal{S}|)f + 4\sum_{p \in R_i^*} d_T(q_p, c_p^*) \right)$$

$$= 2fk^* \log(|\mathcal{S}|) + 4 \sum_{p \in \mathcal{P}_{\text{NEAR}}} d_T(q_p, c_p^*). \qquad \square$$

By combining Lemmas C.1, 3.5, and 3.6 we can finally prove Lemma 3.7.

PROOF OF LEMMA 3.7. First, by Lemma 3.5, it holds

$$\sum_{p \in \mathcal{P}_{\text{NEAR}}} f(p) \leq k^* f + 3 \cdot \sum_{p \in \mathcal{P}} d_T(q_p, \ell_p). \tag{4}$$

However, by Lemma C.1 and Lemma 3.6, we have that

$$\sum_{p \in \mathcal{P}} d_T(q_p, \ell_p) = \sum_{p \in \mathcal{P}_{\text{FAR}}} d_T(q_p, \ell_p) + \sum_{p \in \mathcal{P}_{\text{NEAR}}} d_T(q_p, \ell_p) \leq$$

$$6 \cdot \sum_{p \in \mathcal{P}_{\text{FAR}}} d_T(q_p, c_p^*) + 2\log(|\mathcal{S}|) \cdot k^* f + 4 \cdot \sum_{p \in \mathcal{P}_{\text{NEAR}}} d_T(q_p, c_p^*) = \leq 2\log(|\mathcal{S}|) \cdot k^* f + 6 \cdot \sum_{p \in \mathcal{P}} d_T(q_p, c_p^*).$$

By substituting this bound into (4), we finally get that

$$\sum_{p \in \mathcal{P}_{\text{NEAR}}} f(p) \leq (6\log(|\mathcal{S}|) + 1) \cdot k^* f + 18 \sum_{p \in \mathcal{P}} d_T(q_p, c_p^*). \qquad \square$$

## C.1 Proof of Theorem 3.1

Theorem 3.1 follows simply by combining previous results.

PROOF OF THEOREM 3.1. The cost of the solution computed by TAKEHEED is a random variable that depends on the choice of $T \sim_{\mathcal{D}} \mathcal{T}$ and whose value is $X := \sum_{p \in \mathcal{P}} \left( d(p, \ell_p) + f(p) \right)$.

First, by Lemma 3.2,

$$\sum_{p \in \mathcal{P}} d(p, \ell_p) \leq \sum_{p \in \mathcal{P}} d(p, c_p^*) + \sum_{p \in \mathcal{P}} d_T(q_p, \ell_p).$$

Second, by Lemma 3.4, and Lemmas 3.6 and 3.7,

$$\sum_{p \in \mathcal{P}} \left( d_T(q_p, \ell_p) + f(p) \right) \leq \left( 8\log(|\mathcal{S}|) + 1 \right) fk^* + 34 \sum_{p \in \mathcal{P}} d_T(q_p, c_p^*).$$

Therefore,

$$\mathbb{E}[X] \leq \sum_{p \in \mathcal{P}} d(p, c_p^*) + \left( 8\log(|\mathcal{S}|) + 1 \right) fk^* + 34 \mathbb{E}_{T \sim \mathcal{D}} \Big[ \sum_{p \in \mathcal{P}} d_T(q_p, c_p^*) \Big].$$

Third, by Lemma 3.3,

$$\mathbb{E}_{T \sim \mathcal{D}} \Big[ \sum_{p \in \mathcal{P}} d_T(q_p, c_p^*) \Big] \leq 16 \log(|\mathcal{S}|) \sum_{p \in \mathcal{P}} d(p, c_p^*).$$

By putting everything together, we have

$$\mathbb{E}_{T \sim \mathcal{D}}[X] \quad \leq \quad \left( 8\log(|\mathcal{S}|) + 1 \right) fk^* + \left( 544 \log(|\mathcal{S}|) + 1 \right) \sum_{p \in \mathcal{P}} d(p, c_p^*)$$

$$\leq \quad \left( 544 \log(|\mathcal{S}|) + 1 \right) \text{OPT}(\mathcal{P}, \mathcal{S}) = O\left( \log(|\mathcal{S}|) \, \text{OPT}(\mathcal{P}, \mathcal{S}) \right). \square$$

# D Robustness

Mahdian et al. [2012] showed how to combine two different online algorithms for facility location into a single one whose cost is a constant approximation of the cost of the best algorithm. By applying the same scheme to combine our algorithm TAKEHEED and the randomized algorithm by Meyerson [2001], we obtain an algorithm robust to bad suggestions.

To make our work self-contained, here we report the pseudo-code of the Algorithms by Mahdian et al. [2012], which we call COMBINE, and Meyerson [2001], which we call MEYERSON.

In the Algorithm MEYERSON, for each client, we open a facility at it with probability proportional to its distance to the closest opened facility. After that, we assign the client to its closest opened facility.

---

**Algorithm D.1** Algorithm MEYERSON

1: $\mathcal{F} \leftarrow \emptyset$ set of opened facilities
2: **for** $p \in \mathcal{P}$ **do**
3:     Let $\delta_p := d(p, \mathcal{F})$ (where $d(p, \mathcal{F}) = \infty$ if $\mathcal{F} = \emptyset$)
4:     With probability $\min\left(\frac{\delta_p}{f}, 1\right)$ open a facility at $p$
5:     Assign $p$ to its closest open facility in $\mathcal{F}$

---

The following result, derived by Fotakis [2011], is a slight improvement of the original result of Meyerson [2001].

**Theorem D.1.** *For each input set $\mathcal{P}$, the cost of the solution computed by Algorithm* MEYERSON, *name it* COST(MEYERSON), *satisfies:* $\mathbb{E}[\text{COST}(\text{MEYERSON})] \leq \left(2 + \frac{3 \log(n)}{\log \log(n)}\right) \cdot \text{OPT}(\mathcal{P}) = O\left(\frac{\log(n)}{\log \log(n)} \cdot \text{OPT}(\mathcal{P})\right)$.

---

**Algorithm D.2** Algorithm COMBINE$_\gamma(A_1, A_2)$; input $\gamma > 1$, facility location algorithms $A_1, A_2$

1: **for** $p \in \mathcal{P}$ **do**
2:     **if** COST($A_1$) $\leq (\gamma - 1) \cdot$ COST($A_2$) computed up to this point **then**
3:         Let $q_{A_1}(p)$ be the facility used by Algorithm $A_1$ to serve $p$
4:         **if** $q_{A_1}(p)$ is not open **then** open it
5:         Assign $p$ to $q_{A_1}(p)$
6:     **else**
7:         Let $q_{A_2}(p)$ be the facility used by Algorithm $A_2$ to serve $p$
8:         **if** $q_{A_2}(p)$ is not open **then** open it
9:         Assign $p$ to $q_{A_2}(p)$

---

Algorithm COMBINE is a generic procedure to combine two facility location algorithms. It starts with following the decisions of one of the two algorithms. However, it always switches to the other algorithm when its overall cost, i.e. the cost that would have been paid by following it blindly from the beginning, is "much" smaller than the overall cost of the currently used algorithm. The decision rule for swapping between the algorithms is based on a parameter $\gamma > 1$.

As shown in Mahdian et al. [2012], the following Theorem holds.

**Theorem D.2.** *For a generic input set $\mathcal{P}$, let* COST($A_1$), COST($A_2$), COST(COMBINE$_\gamma(A_1, A_2)$) *be the final cost of the solutions produced respectively by Algorithms $A_1$, $A_2$, and* COMBINE($A_1, A_2$). *They satisfy:* COST(COMBINE$_\gamma(A_1, A_2)$) $\leq \min\left(\frac{\gamma}{\gamma - 1} \cdot \text{COST}(A_1), \gamma \cdot \text{COST}(A_2)\right)$.

We use Algorithm COMBINE on Algorithm MEYERSON and our Algorithm TAKEHEED. By using Theorem D.2 on Algorithm COMBINE(MEYERSON, TAKEHEED), we can finally prove Corollary 3.1.1. Notice that COMBINE(MEYERSON, TAKEHEED) is called WARY in the main paper.

PROOF OF COROLLARY 3.1.1. Consider the Algorithm COMBINE$_\gamma$(MEYERSON, TAKEHEED) with $\gamma = 1 + O(1)$. By Theorem D.2, the expected cost of the resulting Algorithm is bounded by the minimum between:

- $\frac{\gamma}{\gamma-1} \cdot \mathbb{E}[\text{COST}(\text{MEYERSON})]$, which is bounded by $O\left(\frac{\log(n)}{\log\log(n)}\text{OPT}\right)$ by Theorem D.1;

- $\gamma \cdot \mathbb{E}[\text{COST}(\text{TAKEHEED})]$, which is bounded by $O(\log(|\mathcal{S}|)\text{OPT}(\mathcal{P}, \mathcal{S}))$ by Theorem 3.1.

The Corollary follows. □

The robust version of our Algorithm, WARY, is therefore $\text{COMBINE}_\gamma(\text{MEYERSON}, \text{TAKEHEED})$ with a specific value of $\gamma > 1$.

### D.1 Extending Algorithm COMBINE to Multiple Advice

There is a natural extension of Algorithm COMBINE to the setting with multiple advice ($k \geq 2$). Here we will describe this extension, and we will show that its performance can be much worse than the one guaranteed by our Algorithm TAKEHEED.

---

**Algorithm D.3** Algorithm $\text{COMBINE-MUL}_\gamma(A_1, \ldots, A_k)$; input $\gamma > 1$, facility location algorithms $A_1, \ldots, A_k$

---

1: $i \leftarrow 1$
2: **for** $p \in \mathcal{P}$ **do**
3:     **if** $\min_{j\in[k]} \text{COST}(A_j) < (\gamma - 1) \cdot \text{COST}(A_i)$ all computed up to this point **then**
4:         $i \leftarrow \arg\min_{j\in[k]} \text{COST}(A_j)$
5:     Let $q_{A_i}(p)$ be the facility used by Algorithm $A_i$ to serve $p$
6:     **if** $q_{A_i}(p)$ is not opened **then** open it
7:     Assign $p$ to $q_{A_i}(p)$

---

In Algorithm COMBINE-MUL, we start by blindly following one of the Algorithms. However, we always switch to another algorithm when its overall cost, i.e. the cost that would have been paid by following it blindly from the beginning, is "much" smaller than the overall cost of the currently used algorithm. When switching, clearly, we switch to the best-performing algorithm. The decision rule for switching the algorithm is based on a parameter $\gamma > 1$. Algorithm COMBINE-MUL can be used to combine multiple advice. The Algorithm naturally associated with an advice set, i.e. a set of suggested facilities $\mathcal{S}_i$, simply assigns to any client its closest suggested facility in $\mathcal{S}_i$, after opening it if it was not opened yet. Given $\mathcal{S}_1, \ldots, \mathcal{S}_k$ sets of suggested facilities, we will name this combination procedure $\text{COMBINE-MUL}_\gamma(\mathcal{S}_1, \ldots, \mathcal{S}_k)$. With abuse of notation, we will also refer to each $\mathcal{S}_i$ as the described algorithm for the set of suggested facilities $\mathcal{S}_i$.

We will prove a lower bound for the cost computed by $\text{COMBINE-MUL}_\gamma(\mathcal{S}_1, \ldots, \mathcal{S}_k)$. The lower bound shows that Algorithm $\text{COMBINE-MUL}_\gamma(\mathcal{S}_1, \ldots, \mathcal{S}_k)$ can perform much worse than our robust Algorithm TAKEHEED.

**Theorem D.3.** *Let $f = 1$ be the uniform facility cost and $\gamma > 1$ fixed. For every integer $k \geq 2$, there is a finite metric space $(\mathcal{S}, d)$ of cardinality $|\mathcal{S}| = k$ and input sequences such that: (i) The optimal solution has cost $O(1)$, but (ii) Algorithm $\text{COMBINE-MUL}_\gamma(\mathcal{S}_1, \ldots, \mathcal{S}_k)$ computes solutions of cost $\Omega(|\mathcal{S}|) = \Omega(k)$.*

*Proof.* Consider $\mathcal{S}_i = \{1/k^{2i}\}$, $1 \leq i \leq k-1$, and $\mathcal{S}_k = \{0\}$. Let $\mathcal{S}$ be their union. The input set contains the points with multiplicity, where the multiplicities are $m_i := m(1/k^{2i}) = \lceil \gamma \cdot \frac{k^{2i}}{i^2} \rceil$ for $1 \leq i \leq k-1$, $m_k := m(0) = 1$. We will show an $\Omega(k)$-lower bound for Algorithm $\text{COMBINE-MUL}_\gamma(\mathcal{S}_1, \ldots, \mathcal{S}_k)$ on this instance, where the clients are received in this order: $1/k^2, 1/k^4, \ldots, 1/k^{2(k-1)}, 0$.

First, we show that $OPT \leq \pi^2/3 = O(1)$. An upper bound for the cost of the optimal solution is given by the cost of the solution opening only a facility at $0$. This has cost

$$\sum_{i=1}^{k-1}\left(\frac{1}{k^{2i}} \cdot \left\lceil \frac{k^{2i}}{i^2}\right\rceil\right) \leq \sum_{i=1}^{k-1}\left(\frac{1}{k^{2i}} \cdot \left(\frac{k^{2i}}{i^2} + 1\right)\right) \leq \sum_{i=1}^{\infty}\left(\frac{1}{k^{2i}} \cdot \left(\frac{k^{2i}}{i^2} + \frac{k^{2i}}{i^2}\right)\right)$$

because $k^i \geq i$ for each $k \geq 2, i \geq 1$. Thus, the cost is $\leq 2\sum_{i=1}^{\infty}\frac{1}{i^2} = \frac{\pi^2}{3}$ by a well-known result on the sum of the inverse squares of natural numbers.

Second, we show that Algorithm COMBINE-MUL$_\gamma(\mathcal{S}_1, \ldots, \mathcal{S}_k)$ switches to $\mathcal{S}_2, \ldots, \mathcal{S}_\ell$ in this order, where $\ell := \left\lfloor \sqrt{\frac{\gamma-1}{2(2+3\gamma)}}(k-1)\right\rfloor$ (notice that $\ell \leq k-1$), each time opening the facility in $\mathcal{S}_i$, $2 \leq i \leq \ell$. This will imply that it opens $\geq \ell - 1$ facilities, so it produces a solution of cost $\geq \ell - 1 = \Omega(k) = \Omega(|\mathcal{S}|)$. We will show this by induction on $i$. We divide the input in phases $1, \ldots, k$: phase $i$ contains only repetitions of the client in $\mathcal{S}_i$, i.e. $1/k^{2i}$ for $i < k$ and $0$ for $i = k$. It suffices to show that, before the end of each phase $2 \leq i \leq \ell$, it holds: (1) $\text{COST}(\mathcal{S}_i) < \text{COST}(\mathcal{S}_j)$ for each $j \neq i$; (2) $cost(\mathcal{S}_i) \leq (\gamma - 1) \cdot \text{COST}(\mathcal{S}_{i-1})$. Indeed, this will imply a switch from $\mathcal{S}_{i-1}$ to $\mathcal{S}_i$ by the end of each phase $i$. Now, by the inductive hypothesis, before phase $i$ starts, we are using advice $\mathcal{S}_{i-1}$. When half of the points from phase $i$ have been seen, i.e. only $m_i/2$, we have that:

- $\text{COST}(\mathcal{S}_j) > \text{COST}(\mathcal{S}_i)$ for each $j > i$: this is true because each solution pays 1 as facility cost. However, for each client seen until now, its distance to the facility in $\mathcal{S}_i$ is strictly smaller than its distance to the facility in any $\mathcal{S}_j$, $j > i$.

- $\text{COST}(\mathcal{S}_j) > (\gamma-1)\text{COST}(\mathcal{S}_{i-1})$ for each $j < i-1$. This is true by the inductive hypothesis before seeing any point from phase $i$. After that, the points in phase $i$ are farther from $\frac{1}{k^{2j}}$ than $\frac{1}{k^{2(i-1)}}$, so it will still hold $\text{COST}(\mathcal{S}_j) > (\gamma - 1) \cdot \text{COST}(\mathcal{S}_{i-1})$.

- $\text{COST}(\mathcal{S}_i) \leq (\gamma - 1)\text{COST}(\mathcal{S}_{i-1})$. First, $\text{COST}(\mathcal{S}_i) = 1 + \sum_{h=1}^{i-1}\left(\frac{1}{k^{2h}} - \frac{1}{k^{2i}}\right) \cdot m_h + \frac{m_i}{2} \cdot 0 \leq 1 + \gamma \cdot 2\sum_{i=1}^{\infty}\frac{1}{i^2} = 1 + \gamma \cdot \frac{\pi^2}{3} \leq 1 + 4\gamma$. Second, $\text{COST}(\mathcal{S}_{i-1}) \geq 1 + \gamma\sum_{h=1}^{i-1}\left(\frac{1}{k^{2h}} - \frac{1}{k^{2(i-1)}}\right)\frac{k^{2h}}{h^2} + \gamma\left(\frac{1}{k^{2(i-1)}} - \frac{1}{k^{2i}}\right)\frac{k^{2i}}{2i^2} \geq 1 + \gamma\left(\frac{1}{k^{2(i-1)}} - \frac{1}{k^{2i}}\right)\frac{k^{2i}}{2i^2} = 1 + \gamma \cdot \frac{k^2-1}{2i^2}$. Since $i^2 \leq \ell^2 \leq \frac{(\gamma-1)(k-1)^2}{2(2+3\gamma)}$ by definition of $\ell$, we have that $\text{COST}(\mathcal{S}_i) \geq 1 + \gamma \cdot \frac{2(2+3\gamma)(k+1)}{(\gamma-1)(k-1)} \geq 1 + \frac{2\gamma\cdot(2+3\gamma)}{\gamma-1}$. It now suffices to show that $\text{COST}(\mathcal{S}_i) \leq 1 + 4\gamma \leq (\gamma - 1) \cdot \left(1 + \frac{2\gamma(2+3\gamma)}{\gamma-1}\right) \leq (\gamma - 1) \cdot \text{COST}(\mathcal{S}_{i-1})$. It holds $(\gamma - 1) \cdot \left(1 + \frac{2\gamma(2+3\gamma)}{\gamma-1}\right) = 6\gamma^2 + 5\gamma - 1$, and this is $\geq 1 + 4\gamma$ if and only if $6\gamma^2 + \gamma - 2 \geq 0$, which is true because $6\gamma^2 + \gamma - 2 = (2\gamma - 1)(3\gamma + 2) \geq 0$ for $\gamma > 1$.

It follows that Algorithm COMBINE-MUL$_\gamma(\mathcal{S}_1, \ldots, \mathcal{S}_k)$ has cost $\geq \ell - 1 = \Omega(k)$. $\qquad\square$

For the next section, it will also be important to prove an upper bound to the performance of Algorithm COMBINE-MUL. We show that the lower bound of Theorem D.3 is asymptotically tight. The following result is a generalization of the result for Algorithm COMBINE, which is proved in Mahdian et al. [2012].

**Theorem D.4.** *Consider $k$ algorithms $A_1, \ldots, A_k$ and their combination* COMBINE-MUL$_2(A_1, \ldots, A_k)$. *Then, for any set of clients $P$,*

$$\text{COST}(\text{COMBINE-MUL}_2(A_1, \ldots, A_k)) \leq k \cdot \min_{i \in [k]} \text{COST}(A_i).$$

*Proof.* Assume WLOG that COMBINE-MUL has actually chosen the algorithms $A_1, \ldots, A_h$, $h \leq k$, at some point. Let $A_{i^*}$ be the final algorithm used in COMBINE-MUL$_2(A_1, \ldots, A_k)$. By definition, $\text{COST}(A_{i^*}) = \min_{i \in [k]} \text{COST}(A_i)$. Now, we partition the set of clients $P$ into contiguous set of points. Define $P_j \subseteq P$ such that:

- All the points in $P_j$ are contiguous in $P$;

- A single algorithm $A_i$ was used on the points on $P_j$;

- Different algorithms from $A_i$ were used on the points immediately before and after the ones in $P_j$.

We can also define a total order relation among those sets: we say that $P_j < P_j'$ if and only if all the points in $P_j$ are seen before the points in $P_j'$. This relation defines an ordering such that $P_1 < P_2 < \ldots$ and $\bigcup_j P_j = P$. Now, for each $P_j$, we can define $i(j)$ such that $A_{i(j)}$ is the algorithm used for the points in $P_j$. Let $\text{COST}_j$ be the cost paid by algorithm COMBINE-MUL$_2(A_1, \ldots, A_k)$ on all the points seen up to $P_j$ (including the ones in this set), and let $\text{COST}_j(A_{i(j)})$ be the cost paid

by algorithm $A_{i(j)}$ on the same points. We will prove by (complete) induction that $\text{COST}_j \leq |\{i(j') : j' \leq j\}| \cdot \text{COST}_j(A_{i(j)})$.

**Base Case:** $j = 1$. Trivially, we have been using only $A_{i(1)}$. So $\text{COST}_1 = \text{COST}_1(A_{i(1)}) \leq |\{i(1)\}|\text{COST}_1(A_{i(1)})$.

**Inductive Step:** $1, \ldots, (j-1) \to j$. We have used $A_{i(j)}$ for the points in $P_j$. Now, either $i(j) \notin \{i(1), \cdots i(j-1)\}$ or $i(j) \in \{i(1), \cdots i(j-1)\}$.
In the former, We have that $\text{COST}_j \leq \text{COST}(A_{i(j)})|_{P_j} + cost_{j-1} \leq \text{COST}(A_{i(j)})|_{P_j} + |\{i(1), \cdots i(j-1)\}|cost_{j-1}(A_{i(j-1)})$ by the inductive hypothesis. However, since we were not using $A_{i(j)}$ at the previous step, but $A_{i(j-1)}$, it holds $cost_{j-1}(A_{i(j-1)}) \leq cost_{j-1}(A_{i(j)})$. Thus, $\text{COST}_j \leq \text{COST}(A_{i(j)})|_{P_j} + (|\{i(1), \cdots i(j)\}| - 1)cost_{j-1}(A_{i(j)}) = |\{i(j') : j' \leq j\}| \cdot \text{COST}_j(A_{i(j)})$.
In the latter, we can treat separately the clients managed by $A_{i(j)}$ across time and the other clients, for which we can still use the inductive hypothesis with respect to the algorithms $\{A_{i(j')} : j' < j\} \backslash \{i(j)\}$. It is equivalent to the previous case in which all the points managed by $A_{i(j)}$ are merged directly into $P_j$. Thus, analogously, we get the desired bound.

By applying the proved inequality to the last set $P_{i^*}$, we get that

$$\text{COST}(\text{COMBINE-MUL}_2(A_1, \ldots, A_k)) \leq h \cdot \text{COST}(A_{i^*}) \leq k \cdot \min_{i \in [k]} \text{COST}(A_i).$$

$\square$

# E Additional Robustness to Bad Advice Sets: BUCKETHEED

Our Algorithm TAKEHEED could be tricked by just a single bad advice set: if there is an advice set, name it $\mathcal{S}_{bad}$, which has a much larger cardinality than the optimal advice and a much larger cost, Theorem 3.1 shows that we could get a $\log(|\mathcal{S}_{bad}|)$ approximation of the desired solution. If $|\mathcal{S}_{bad}| = \Theta(n)$, we could get worse guarantees than MEYERSON Algorithm, which has no access to any advice. This observation suggests that TAKEHEED could not be robust to the presence of a few bad advice sets, even if the other advice sets are very good, and motivates our proposal of a more robust algorithm, BUCKETHEED.

Here we describe Algorithm BUCKETHEED and prove theoretical guarantees for its performance. We do not perform an experimental analysis on it because, in practice, advice sets are *not* adversarial. In our experiments, we show that we can always guarantee to have the advice of roughly the same cardinality. The motivation behind the introduction of BUCKETHEED is purely theoretical and further validates our approach.

Let $\Delta := \max_{i \in [k]} |\mathcal{S}_i|$. We assume to have a polynomial approximation, $\tilde{n}$, of the number of clients $n$ (hence $\log n = \Theta(\log \tilde{n})$). Then, we can assume that $\log(\Delta) = O(\log(n))$ (i.e., $\Delta \in poly(n)$): a set with more than $n$ facilities can be discarded, because it always has a larger cost than the trivial algorithm which opens a facility at each client. Moreover, we can suppose that $\log(k) = O(\log(n))$ as well. If this is not true, Theorem 4.1 shows that we cannot get a better approximation ratio than MEYERSON Algorithm.

## E.1 Algorithm BUCKETHEED

The idea behind Algorithm BUCKETHEED is to divide the advice sets into $O(\log \log(n))$ advice "buckets" that contain advice sets with limited cardinality. We then use Algorithm COMBINE-MUL to combine applications of Algorithm TAKEHEED on the single buckets. By an appropriate definition of the buckets, we make sure that the cost of following Algorithm TAKEHEED on a single bucket, is a $\sim \log(k)$ application of the best advice set in that bucket. Moreover, since the buckets are only $O(\log \log(n))$, the overall algorithm will achieve a $\sim O(\log \log(n) \cdot \log(k))$ approximation of the best oracle, which is closer to the $\Omega\left(\frac{\log(k)}{\log \log(k)}\right)$ approximation lower bound from Theorem 4.1.

We now divide the advice sets into non-disjoint buckets $\{\mathcal{B}_i\}_{i \in [h]}$.

**Definition E.1.** *Consider the advice sets $\{\mathcal{S}_j\}_{j \in [k]}$. Let the buckets $\mathcal{B}_0 := \{\mathcal{S}_j, j \in [k] : |\mathcal{S}_j| = 1\}$ and $\mathcal{B}_i := \{\mathcal{S}_j, j \in [k] : 2^{2^{i-1}} \leq |\mathcal{S}_j| < 2^{2^i}\}$ for each $i \geq 1$. Let $h \geq 0$ be the minimum value such*

*that $\mathcal{B}_h$ contains all the advice sets. Then, we consider as buckets*

$$\mathcal{B}_0, \mathcal{B}_1, \ldots, \mathcal{B}_h.$$

Notice that, by definition,

$$h \leq \log \log \left( \left| \bigcup_{j \in [k]} \mathcal{S}_j \right| \right) \leq \log \log(k\Delta) = O(\log \log(n)).$$

This is true because of our assumption on $k, \Delta$.

We can now define BUCKETHEED.

**Definition E.2.** BUCKETHEED:=COMBINE-MUL$_2(A_1, \ldots, A_k)$, *where, for each $i \in [k]$, $A_i$ :=TAKEHEED$\left( \bigcup_{\mathcal{S}_j \in \mathcal{B}_i} \mathcal{S}_j \right)$.*

We also report the full pseudocode of Algorithm BUCKETHEED.

---

**Algorithm E.4** Algorithm BUCKETHEED$(\mathcal{S}_1, \ldots, \mathcal{S}_k)$. Input: advice sets $\mathcal{S}_1, \ldots, \mathcal{S}_k$

---

1: $h \leftarrow \lceil \log \log(\max_{i \in [k]} |\mathcal{S}_i|) \rceil$
2: $\mathcal{B}_j \leftarrow \emptyset \ \forall \ 0 \leq j \leq h$
3: **for** $i = 1, \ldots, k$ **do**
4:    **if** $|\mathcal{S}_i| = 1$ **then**
5:       $\mathcal{B}_0 \leftarrow \mathcal{B}_0 \cup \{\mathcal{S}_i\}$
6:    **else**
7:       **for** $j = 1, \ldots, h$ **do**
8:          **if** $2^{2^{i-1}} \leq |\mathcal{S}_i| < 2^{2^i}$ **then**
9:             $\mathcal{B}_j \leftarrow \mathcal{B}_j \cup \{\mathcal{S}_i\}$
10: $i \leftarrow 1$
11: **for** $p \in \mathcal{P}$ **do**
12:    **if** $\min_{j \in [h]} \text{COST}(\text{TAKEHEED}(\mathcal{B}_j)) < \text{COST}(\text{TAKEHEED}(\mathcal{B}_i))$ all computed up to this point **then**
13:       $i \leftarrow \arg\min_{j \in [h]} \text{COST}(\text{TAKEHEED}(\mathcal{B}_j))$
14:    Let $q_i(p)$ be the facility used by Algorithm TAKEHEED$(\mathcal{B}_i)$ to serve $p$
15:    **if** $q_i(p)$ is not opened **then** open it
16:    Assign $p$ to $q_i(p)$

---

We will show the following robustness result for BUCKETHEED.

**Theorem E.1.** *Consider an advice set $\{\mathcal{S}_j\}_{j \in [k]}$. Let $S_j^* := \arg\min_{\mathcal{S}_j} \text{COST}(\mathcal{S}_j)$ be a best advice set for a specific input set. Then,*

$$\mathbb{E}[\text{COST}(\text{BUCKETHEED})] \leq O\left(\log \log(n) \cdot (\log(k) + \log(|S_j^*|))\right) \cdot \text{COST}(S_j^*).$$

*Proof.* Let $\mathcal{B}_{i^*}$ be the bucket containing $S_j^*$. By Theorem D.4, we have that $\mathbb{E}[\text{COST}(\text{BUCKETHEED})] \leq (h + 1) \cdot \mathbb{E}[\text{COST}(A_j)]$. However, we have noticed that $h = O(\log \log(n))$. Moreover, by Theorem 3.1, $\mathbb{E}[\text{COST}(A_j)] \leq O(\log(|\mathcal{B}_{i^*}|)) \cdot \text{COST}(S_j^*)$. Now, by definition of bucket, either $|S_j^*| = 1$, or $i^* \geq 1$ is such that $2^{2^{i^*-1}} \leq |S_j^*| < 2^{2^{i^*}}$. In the former, we are done because $\log(|\mathcal{B}_{i^*}|) \leq \log(k)$. In the latter,

$$|\mathcal{B}_{i^*}| < k \cdot 2^{2^{i^*}} = k \cdot \left(2^{2^{i^*-1}}\right)^2 \leq k \cdot |S_j^*|^2.$$

This implies that

$$\log(|\mathcal{B}_{i^*}|) \leq \log(k) + 2\log(|S_j^*|).$$

By putting everything together, we get the desired bound. $\qquad\square$

Theorem E.2 entails that Algorithm BUCKETHEED is robust to the presence of very large, bad, advice sets, since they do not affect the resulting bound. This theorem, then, further validates our HST-based approach.

# F   Lower Bounds: the Analysis

PROOF OF THEOREM 4.1.  Consider a full binary tree of height $H = 2^t$, for an integer $t \geq 1$. Let $L$ be its set of leaves; then, $|L| = 2^H = 2^{2^t}$.

Given $v \in L$, we define $a_0(v) = v$, and we define $a_{i+1}(v)$ to be the parent of $a_i(v)$.

Given $v \in L$, and $0 \leq i \leq H$, let $L_i(v) \subseteq L$ be the set of leaves that are descendants of $a_i(v)$. Thus, $L_0(v) = \{v\}$, and $L_H(v) = V$.

Given two leaves $\{v, w\} \in \binom{V}{2}$, let $h(v, w)$ be the height of the smallest subtree containing both $v$ and $w$ — for instance, if $v$ and $w$ have the same parent, $h(v, w) = 1$; if the least common ancestor of $v$ and $w$ is the root of the tree, then $h(v, w) = H$.

The metric space on $V$ is defined as follows.  For $\{v, w\} \in \binom{V}{2}$, the distance $d(v, w)$ is equal to $d(v, w) = 2^{h(v,w)-H}$.  In particular, if $\{v, w\} \in \binom{L_{H-i}(v)}{2}$, then $d(v, w) \leq 2^{-i}$.  We set the cost of opening a facility to be $f = 1$.

At the outset, the adversary will choose a node $s \in V$.  The adversary will then pick the sequence of facilities independently of the algorithm to be tested: for each phase $p = 0, \ldots, H$, and for each $i = 1, 2, \ldots, \lceil 2^p/H \rceil$, the adversary will pick a uniform-at-random client $c_{p,i}$ in $L_{H-p}(s)$.  Observe that $d(c_{p,i}, s) \leq 2^{-p}$, and $d(c_{H,i}, s) = 0$.

First, observe that the algorithm that opens $s$ at the outset, and uses it to serve all the clients, will incur in a total cost of at most

$$
\begin{aligned}
1 + \sum_{p=0}^{H-1} \sum_{i=1}^{\lceil 2^p/H \rceil} d(s, c_{p,i}) &\leq 1 + \sum_{p=0}^{H-1} \sum_{i=1}^{\lceil 2^p/H \rceil} 2^{-p} \\
&= 1 + \sum_{p=0}^{H-1} \left( \left\lceil \frac{2^p}{H} \right\rceil \cdot 2^{-p} \right) \leq 1 + \sum_{p=0}^{H-1} \left( \left( \frac{2^p}{H} + 1 \right) \cdot 2^{-p} \right) \\
&\leq 1 + \sum_{p=0}^{H-1} \left( \frac{1}{H} + 2^{-p} \right) < 4.
\end{aligned}
$$

That is, there exists a leaf that, if opened and used to serve each client, acts as a solution having a total cost of $O(1)$.

Now, create one oracle for each leaf $v \in L$.  The $v$-oracle, $\mathcal{S}_v$ will suggest to open a single facility: $v$.  Then, the best oracle will incur in a total cost of $O(1)$.  Moreover, the number of oracles equals $k = |L| = 2^H$, the set of all suggested facilities is $\mathcal{S} = L$ and has cardinality $k = 2^H$.

Then, observe that if an algorithm opens more than $0.1H/\lg(H)$ facilities, its total cost will be larger than the cost of that of the best oracle by a factor of $\Omega(H/\log(H)) = \Omega(\log(k)/\log\log(k))$.

In the remainder of the proof, we will then only consider algorithms opening no more than $0.1H/\lg(H) = O(\log(k)/\log\log(k))$ facilities.

Suppose that the adversary picks the node $s$ uniformly at random.  From the algorithm's perspective, this amounts to the following distributions on the facilities.  For each $p = 1, \ldots, H$, the $p$th batch will contain $\lceil 2^p/H \rceil$ clients, each of which will be chosen independently and uniformly at random in a set $M_p$, that the adversary known to be equal to $M_p = L_{H-p}(s)$, and that, from the algorithm's perspective during phase $p-1$, will be chosen uniformly at random in the class $C_p = \{M_p, L_{H-p+1}(s) - M_p\}$.  The algorithm, once phase $p-1$ begins, knows the class $C_p$, but will have no way of inferring, before phase $p$ begins, which of its two sets will end up being the support of the distribution of phase $p$.  (Indeed, the distribution of the clients up to and including those in phase $p-1$ is independent of the uniform-at-random bit choosing the support of the distribution of phase $p$.)  Analogously, at phase $p-1$, the algorithm only knows that the set $M_{p+2\ell}$ will be chosen UAR from a class of $2^{2\ell+1}$ sets.

Now, suppose that the algorithm does not open facilities in any of the phases $p+1, \ldots, p+2\ell$.  Consider any of the previously opened facilities $f$, and any client $c$ of phase $p+2\ell$.  Let $\xi$ be the event that $d(c, f) \leq 2^{-p-\ell}$.  For $\xi$ to happen, it must be that $c$ is a leaf of the set $L_{H-p-\ell}$ — since, at phase

$p, c$ is uniformly distributed in (a superset of) $L_{H-p}$, we have that $\Pr[\xi] \leq O\left(\frac{|L_{H-p-\ell}|}{|L_{H-p}|}\right) = O\left(2^{-\ell}\right)$. Since the number of facilities opened by the algorithm is at most $O(H/\log H)$, by the union bound, the probability that there exists a facility opened by the algorithm before phase $p+1$ that is at distance at most $2^{-p-\ell}$ from $c$ is at most $O(2^{-\ell}H/\log H)$.

If we let $\ell = \lg(H)$, this probability becomes at most $O(1/\log H)$. That is, the expected cost for serving $c$, if no facilities were opened in phases $p+1, \ldots, p+2\ell$, is at least $\Omega(2^{-p-\ell})$ — then, since phase $p+2\ell$ contains at least $2^{p+2\ell}/H$ clients, the expected cost for phase $p+2\ell$, with no facilities opened in phases $p+1, \ldots, p+2\ell$, is at least $\Omega(2^{-p-\ell} \cdot 2^{p+2\ell}/H) = \Omega(2^{\ell}/H) = \Omega(1)$.

Now, if the algorithm has opened at most $0.1H/\lg H$ facilities, then there must exist at least $H/2$ phases $p$ such that the algorithm has not opened any facilities in the phases $p-2\ell, p-2\ell+1 \ldots, p-1, p$ (indeed, each opened facility can cover at most $2\ell = 2\lg(H)$ phases and, thus, the number of covered phases is at most $0.2H$, less than half of the total). Therefore, if the algorithm has opened at most $0.1H/\lg(H)$ facilities, its expected (distance) cost is at least $\Omega(H) = \Omega(\log k)$ — a $\Omega(\log k)$ factor larger than the cost of the best oracle. $\square$

PROOF OF THEOREM 4.2. Consider $\mathcal{S} = \{1/2^{2i}, \ 1 \leq i \leq k-1\} \cup \{0\}$. The set of clients contains the points in this order with multiplicity, where the multiplicities are $m_i := m(1/2^{2i}) = \lceil \frac{2^{2i}}{i^2} \rceil$ for $1 \leq i \leq k-1$, $m_k := m(0) = 1$. We will show an $\Omega(k)-$lower bound for any Meyerson-like algorithm on this instance, where the clients are received in this order: $1/2^2, 1/2^4, \ldots, 1/2^{2(k-1)}, 0$.

First, we show that $OPT \leq \pi^2/3 = O(1)$. An upper bound for the cost of the optimal solution is given by the cost of the solution opening only a facility at $0$. This has cost

$$\sum_{i=1}^{k-1} \left(\frac{1}{2^{2i}} \cdot \left\lceil \frac{2^{2i}}{i^2} \right\rceil\right) \leq \sum_{i=1}^{k-1} \left(\frac{1}{2^{2i}} \cdot \left(\frac{2^{2i}}{i^2} + 1\right)\right) \leq \sum_{i=1}^{\infty} \left(\frac{1}{2^{2i}} \cdot \left(\frac{2^{2i}}{i^2} + \frac{2^{2i}}{i^2}\right)\right)$$

because $2^i \geq i$ for each $i \geq 1$. Thus, the cost is $\leq 2\sum_{i=1}^{\infty} \frac{1}{i^2} = \frac{\pi^2}{3}$ by a well-known result on the sum of the inverse squares of natural numbers.

Second, any Meyerson-like algorithm pays cost $\geq 1$ on each location in $\mathcal{S}$ except $0$. We will show this by induction on $i$. By definition of Meyerson-like algorithm, a facility is opened at the first input point ($i = 1$), with resulting cost paid $= 1$. Now, when the $i-$th distinct input point is seen, its distance to its closest previous input point, and so to the closest opened facility, is $\geq \frac{1}{k^{2(i-1)}} - \frac{1}{k^{2i}}$, and its multiplicity is $\lceil \frac{k^{2i}}{i^2} \rceil$. If we, sooner or later, open a facility at this location, we are done since we pay $1$ to do that. Assume this is not true, so for points in this location we will pay $\geq \left(\frac{1}{k^{2(i-1)}} - \frac{1}{k^{2i}}\right) \cdot \frac{k^{2i}}{i^2} = \frac{k^2-1}{i^2} = \left(\frac{k}{i}\right)^2 - \frac{1}{i^2} \geq \frac{k}{i} - \frac{1}{i} \geq 1$ because $i \leq k-1$.

Third, for this specific class of instances SNFL algorithm from Fotakis [2005] behaves exactly as a Meyerson-like algorithm, so it produces a solution with cost $\geq k-1 = \Omega(k)$. We also show this by induction on the different locations of the input points. In the inductive hypothesis, we also include a fact about possible facilities that have not been touched by any input point: they could not be opened facilities, and their potentials are smaller the closer they are to $0$. By definition of SNFL algorithm, a facility is opened at the first input point ($i = 1$), exactly as for Meyerson-like algorithms, and nothing is done for the other points. Now, when the $i-$th distinct input point is seen, its distance to its closest previous input point, and so to the closest opened facility by inductive hypothesis, is $\geq \frac{1}{k^{2(i-1)}} - \frac{1}{k^{2i}}$, and its multiplicity is $\lceil \frac{k^{2i}}{i^2} \rceil$. By inductive hypothesis, there is not an opened facility at $\frac{1}{k^{2i}}$ yet. However, its potential is not smaller than the potentials of the possible facilities that have not been opened yet, because those as closer to $0$. Subsequent arrivals of input points at $\frac{1}{k^{2i}}$ may increase the potentials of all the closed possible facilities, but the increases are always smaller for closer points to $0$. At a certain point, for what shown for Meyerson-like algorithms, the potential of $\frac{1}{k^{2i}}$ reaches $1$, so it gets opened as a facility (recall that it is the largest potential of the remaining closed possible facilities). For the remaining input points at $\frac{1}{k^{2i}}$, no potentials are touched and we are done. $\square$

# G Experimental Results

In this section, we report the experimental results for *Gowalla* and *Uber*. The behavior is similar to *Brightkite*, reported in the experimental section of the main paper. We include plots showing the performance of the robust variant of our algorithm with different values of the mixing parameter $\gamma$ (Figures 5, 9, 13). For completeness, we also show the quality of each advice of the set supplied to the advice-based algorithms (Figures 6, 10, 14). Finally, we included a plot showing the running time of the different algorithms (Figure 15).

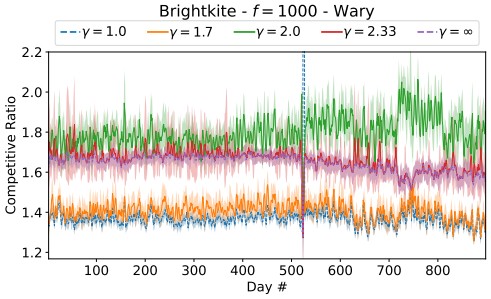

Figure 5: Average estimated competitive ratios for different mixing parameter $\gamma$ between TAKEHEED and MEYERSON for robustness (by $\gamma = 1$ we mean the non-robust version). The shaded area represents one standard deviation.

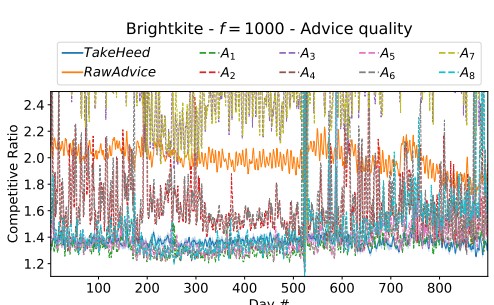

Figure 6: Average estimated competitive ratios for the single advice sets and the algorithms combining them. The shaded area represents one standard deviation.

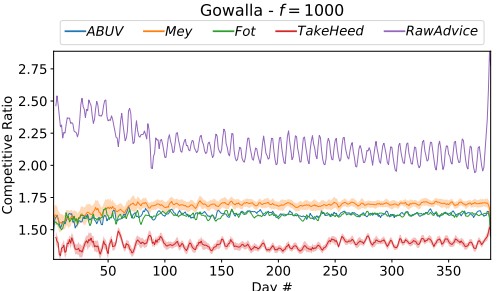

Figure 7: Average estimated competitive ratios of the main online algorithms. The shaded area represents one standard deviation.

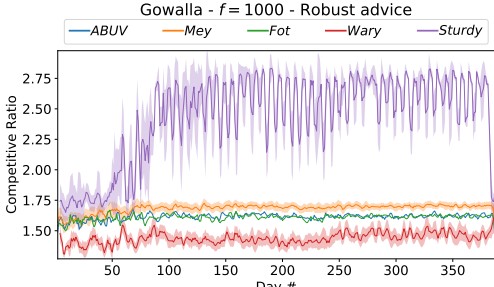

Figure 8: Average estimated competitive ratios of the robust version of main online algorithms. The shaded area represents one standard deviation.

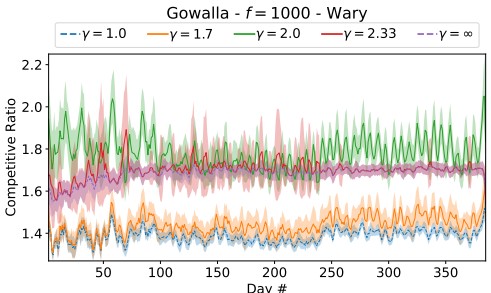

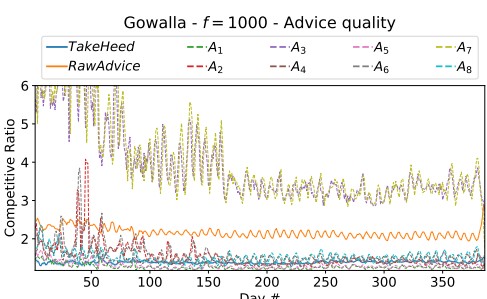

Figure 9: Average estimated competitive ratios for different mixing parameter $\gamma$ between TAKEHEED and MEYERSON for robustness (by $\gamma = 1$ we mean the non-robust version). The shaded area represents one standard deviation.

Figure 10: Average estimated competitive ratios for the single advice sets and the algorithms combining them. The shaded area represents one standard deviation.

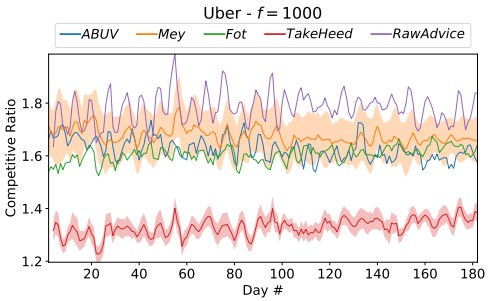

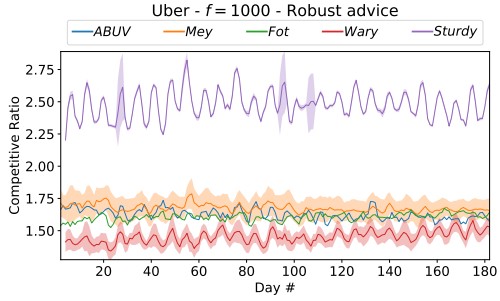

Figure 11: Average estimated competitive ratios of the main online algorithms. The shaded area represents one standard deviation.

Figure 12: Average estimated competitive ratios of the robust version of main online algorithms. The shaded area represents one standard deviation.

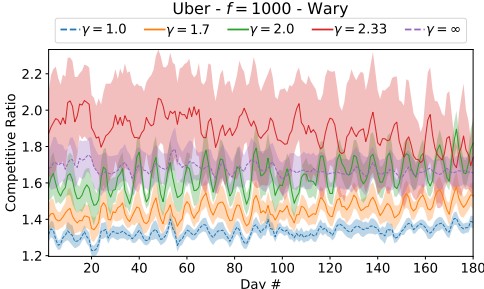

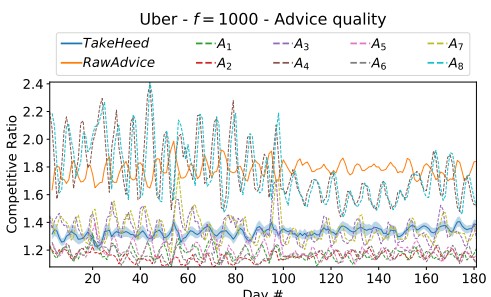

Figure 13: Average estimated competitive ratios for different mixing parameter $\gamma$ between TAKEHEED and MEYERSON for robustness (by $\gamma = 1$ we mean the non-robust version). The shaded area represents one standard deviation.

Figure 14: Average estimated competitive ratios for the single advice sets and the algorithms combining them. The shaded area represents one standard deviation.

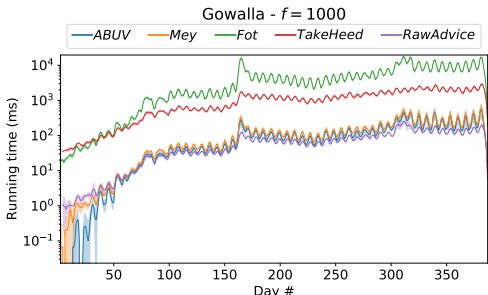

Figure 15: Average running time of the main online algorithms to process all the clients. For TAKEHEED, it includes the time needed to sample the HST. The shaded area represents one standard deviation.