# OpenReview forum: "Online Facility Location with Multiple Advice"
_NeurIPS.cc/2021/Conference — NeurIPS 2021 Poster_

### Official Review · Reviewer_ueSN · 2021-07-15

**Rating:** 8
**Confidence:** 4

**Summary:**

The paper studies the problem of online (uniform) facility location with multiple expert advices. Here the expert advices are given in the form of the offline optimal facility set: i.e., each advice is a set of facilities that are claimed to be open by OPT. The main result of the paper is an algorithm with cost $O(\log(|S|))\mathrm{OPT}(S)$, where $S$ is the *union* of all the expert advices, and $\mathrm{OPT}(S)$ is the best (offline) cost one may get by using only facilities from $S$. By combining with known worst-case online algorithms (e.g. Meyerson's algorithm) using standard blackbox method, one easily get an algorithm with robustness-consistency performance guarantee.

The main idea of the $O(\log|S|)\mathrm{OPT}(S)$-cost algorithm is to first embed the facility set to an HST, then run an alg similar to Meyerson/Fotakis, except for one important difference: when the algorithm decides to open a facility in some region, it (Algorithm 3) choose to open the "limit point" in that region rather than the facility giving smaller cost. In particular, this difference enables the algorithm to circumvent the classical lower bound instance given by [Meyerson'01]: the instance consists of a sequence of clients that converges to a point in the space.

Finally, notice that since the goal is to approximate $\mathrm{OPT}(S)$, one can w.l.o.g. move all clients to the location of $S$ losing only an additional $O(1)$ factor. Then using similar argument as in [Fotakis'11] one can get the $O(\log|S|)*\mathrm{OPT}(S)$ cost, which could be better than the $O(\log n)$-apx of Meyerson/Fotakis when $\mathrm{OPT}(S)$ is close to OPT.

**Main Review:**

**[Strength]**
I think the main contribution is the $O(\log|S|)*\mathrm{OPT}(S)$-cost algorithm, while the rest part is rather standard. I like the idea of using HST to identify the limit point in (a subset of) the facilities. Simple but clever. The result is also interesting even without the learning-augmented background: it implies better competitive ratio when the facilities come from a fixed set (that's much smaller than potential client sets), which is often the case in practice. The authors also provide a lower bound (almost) matched by their algorithm. I don't have time to look at the experiments, but I think it's not important for the evaluation of this paper.

**[Weakness]**
Although the lower bound Theorem 4.1 is incomparable with that given by [Fotakis'03], it is not hard to obtain by a small modification. In fact the lower bound construction given in the paper is almost identical with that of [Fotakis03].
Finally, the author should discuss a little bit more on the primal-dual algorithm by [Fotakis'05], which is not "Meyerson-like" and thus Theorem 4.2 does not directly apply.

**Time Spent Reviewing:**

6

---

> ### Author Response · Authors · 2021-08-05
> **Response to Reviewer ueSN**
>
> We thank you for your suggestions.
>
> We agree that the primal-dual algorithm by Fotakis (2005), named SNFL, is not Meyerson-like. However, the same counterexample used for Meyerson-like algorithms works: SNFL is forced to open the same facilities opened by a Meyerson-like algorithm, resulting in an exponentially larger cost than TakeHeed. We thank you for this interesting observation: we will integrate the result in the final version of the paper.

---

### Official Review · Reviewer_b1z8 · 2021-07-15

**Rating:** 7
**Confidence:** 4

**Summary:**

The paper studies the online facility location problem in a setting where you have multiple advice in the form of a set of lists of suggested facilities to open. They introduce an algorithm that they show to competes with the best possible algorithm for online facility location even when the advice is bad and outperforms the best possible when the advice is good.

**Ethical Concerns:**

No major ethical concerns as far as I can see.

**Limitations And Societal Impact:**

The advice setting in the paper while being adversarial, is not allowed to change after the initial selection though this is simply the framework considered for the scope of the paper. Also, the bounds in the paper are derived against the optimal solution restricted to $S$ and it is not clear how it compares against the true optimal. The concept of bad advice seems to be loosely defined.

As the problem itself is a classical problem I don’t think it creates any major negative societal impacts.

**Main Review:**

The paper is well written and is well structured. The arguments seem to follow clearly and the structure of the arguments is clearly laid out. The paper uses the structural properties of HSTs to design and analyze a robust framework for the online facility location with advice. It seems that the advice in this setting could be adversarial. But since the advice does not change over time, the adversary is non-adaptive. I know this is out of the scope of the paper, but is it possible to say anything about what might happen if you are allowed to change some $\epsilon/n$ fraction of the advice at each time? Could this be counted towards bad advice and the robustness arguments used to say that even in this case the performance is good? Feel free to skip this question if you have to, as it is out of the scope. Also, what is the significance of Wary against Robust-Takeheed, if the best possible advice set is also an adversarial set that is large (something like $\epsilon n /k$) how would the two approaches compare? Also, how does $OPT(P,S)$ compare against the true optimal (could this cause any additional $S$ factors especially pertaining to the lower bound analysis)? Also, is there a formal definition of bad advice?

On the formatting front, it might be nice to have the pluck algorithm in a single column as double columns make it hard to compare some structural properties, i.e. indentation making it slightly harder to read. This is just a suggestion as it is not a major issue anyway.

**Time Spent Reviewing:**

2

---

> ### Author Response · Authors · 2021-08-05
> **Response to Reviewer b1z8**
>
> We thank you for your suggestions.
>
> First, we notice that our algorithm is specifically designed to deal with a set of suggested facilities provided in advance. In the beginning, we sample an HST and it cannot be updated. We cannot sample a different HST for each set of facilities, because our algorithm relies on a node potential function that needs to be preserved at each time step. Therefore, we can say that our algorithm is not trivially generalizable to the setting in which suggested facilities change over time, although this is a very interesting question for future works.
>
> Second, $OPT(P,S)$ could be much larger than the offline optimum $OPT$. This is why we need Mahdian mixing technique to derive Algorithm Wary, which always achieves an $O(\log(n)/\log\log(n)) $ approximation of $OPT$.
>
> Finally, there is not a formal definition of bad advice in the paper. However, we could deem an advice set as “bad” if, by following it blindly, we achieve a cost of $\Omega(\log(n)/\log\log(n) * OPT)$. We will add this to the paper.
>
> We highly appreciate your concerns about formatting, and we will address them in the final version of the paper.

---

### Official Review · Reviewer_GcD7 · 2021-07-16

**Rating:** 6
**Confidence:** 4

**Summary:**

The paper looks at the problem of online facility location with multiple advice.

In the online facility location problem points of a metric space arrive online and once a point arrives it either has to be connected to an already open facility, or a new facility gets opened to serve it. These decisions are irrevocable and the goal is to minimise the total cost which consists of the cost for opening the set of facilities and the sum of distances from each point to the facility that serves it. The best known algorithm for the problem is due to Fotakis and obtains a competitive ratio of $O(\log(n)/\log\log(n))$.

The current paper assumes that the algorithm has obtained $k$ different sets of points as "advice". Each such set of points $S_i$ can be thought to represent a set of suggested facilities to open, but it is unclear which -- if any -- of these sets lead to a good solution and which not. The paper uses this advice to develop algorithm TakeHeed with expected cost $O(\log|S|)\cdot OPT(S)$, where $S$ is the union of advice sets and $OPT(S)$ the best solution among the ones that are restricted to use facilities of $S$. Clearly, $OPT(S)$ cannot be worse than the optimal solution restricted to each advice set, so intuitively the algorithm is good when there are relatively few points in the advice sets and at least one of them is reasonably accurate. The algorithm can be robustified by standard techniques (Mahdian et al.) and an additional algorithm is given in the supplementary material for the (arguably not so natural case) in which $|S|$ is large. This additional algorithms compares directly to the quality of the best advice set.

The paper also contains expermints both on synthetic and real world (albeit from different settings) inputs.

**Ethical Concerns:**

No ethical concerns

**Limitations And Societal Impact:**

no limitations or social impact concerns.

**Main Review:**

Algorithm TakeHeed builds upon hierarchically separable trees and, although quite involved and non-trivial, is elegant at the same time. Furthermore the paper is really well written and the clustering problem that is studied is very interesting.

That being said, I have two concerns regarding the paper:

1. I am not fully convinced with the multiple predictions model. I would consider it more natural to have one predicted set of facilities to open and to give the competitive ratio as a function of the error (some distance measure between the predicted set and the actual optimal set of facilities). The choice of multiple predictions model seems also a bit strange given that, at least as far as I can tell, it is the union of the predicted sets that is used in the algorithm (and on top of that, that also appears in the performance guarantee of the algorithm). Although the additional algorithm actually works with individual prediction sets it is again not clear how to compare it to the best algorithm without predictions (see concern (2.) below).

2. the performance of the algorithm is given with respect to cost of the algorithm and not with respect to the competitive ratio which makes it quite hard to compare the theoretical result to other algorithms. In particular the given costs may depend among other things on the number of facilities in a predicted or optimal solution.

Finally, the experimental results are quite interesting in that they (i) support the belief that such multiple advice sets can indeed be obtained from historical data, and (ii) show that the algorithm actually outperforms other algorithms. So although I think that the prediction model is not the most natural, and the theoretical results are maybe not as meaningful to interpret as a performance guarantee, it seems that at least in practice the algorithm performs reasonably well.

I think that the paper is slightly above borderline but would be more than happy to reconsider my score after the rebuttal of the paper.

Some minor notes for the authors:
- The algorithm naming is a bit unfortunate, in particular since the robustified version of TakeHeed is called Wary, but a different algorithm is called Robust-TakeHeed.
- On line 62 you talk about the performance guarantee, but what you write is actually the cost of the algorithm.

**Time Spent Reviewing:**

3

---

> ### Author Response · Authors · 2021-08-05
> **Response to Reviewer GcD7**
>
> We thank you for your suggestions.
>
> We try to address your main concerns.
> 1. Although your suggestion about using a different error measure seems very close to standard notions of error in this setting, we believe it is difficult to use in practice. First, there may be multiple optimal sets of facilities, depending on the symmetries of the input set of points. This would make it difficult to compare the set of suggested facilities to an optimal one. Even if we fix a strategy to choose the optimal solution to compare against, we believe that the error measure should take into account the service cost too. In the facility location problem, the goal of each algorithm is to find somehow a balance between the facility cost and the service cost: avoid opening too many facilities and, at the same time, avoid paying too much service cost. Moreover, one facility could serve many more clients than another facility, so an error for that facility needs to be weighted more. Therefore, the cost of the problem, which takes into account both the facility cost and the service cost, may be a very good “error measure” for the set of predicted facilities. However, we think that your suggestion is interesting and we will add it as an open problem.
> * In addition, we study the multiple advice setting because it is a more difficult and general case. Furthermore, in our experimental section, we show that in practice it is useful to consider multiple advice. Finally, note that even if in our algorithm we take the union of all the sets or a carefully chosen subset of them (as in Robust-TakeHeed), in our analysis we can still give guarantees with respect to the best possible advice (as in the guarantees of Robust-TakeHeed).
> 2. We are trying to provide a solution whose cost is close to the cost of the optimal solution and, at the same time, close to the cost of a suggested solution if it is of good quality. Algorithm TakeHeed exploits the advice effectively. However, the advice could be inaccurate. This is why we introduce Algorithm Wary: by combining TakeHeed and Meyerson’s Algorithm through Mahdian mixing procedure, it achieves both an $O(\log(n)/\log\log(n))$ approximation of the offline optimum and a good approximation of the cost of the suggested solution. This makes our Algorithm comparable to all the main algorithms for the same problem. Moreover, if we receive a good quality suggested solution, we can exploit it effectively and break the logarithmic approximation barrier.
>
> Point well taken concerning the names of the algorithms! Perhaps Robust-TakeHeed can become BucketHeed?

---

### Official Review · Reviewer_gYHr · 2021-07-19

**Rating:** 5
**Confidence:** 4

**Summary:**

This paper is about online algorithms augmented with predictions of the future input. In particular, the authors examine the problem of online facility location. In the classical version of this problem, there is a sequence of points in a metric space that arrive in an online fashion and represent the location of a client. The algorithm has to irrevocably decide to either connect each of those clients to an existing facility paying some “service cost” proportional to the distance from it, or a new one that would incur an additional “facility cost” $f$ to be created.
The authors here consider a version of the problem where the algorithm is provided with advice from multiple sources. Each piece of advice $S_i$ consists of a set of points where the optimal locations of the facilities are predicted to be. The first algorithm proposed uses the union of those predictions $\mathcal{S}=\cup_i \{S_i\}$ and incurs a cost which scales proportionally to $\log|\mathcal{S}|$ and the optimal cost given that only facilities in $\mathcal{S}$ can be used.
The algorithm and its proof is quite technical and makes use of hierarchically separated trees, that had been also used in the classical version of this problem. Matching lower bounds up to a $\log\log |\mathcal{S}|$ factor are also provided as well as experimental results to compare with prior work.


**Limitations And Societal Impact:**

The authors as well as myself do not see any potential negative societal impact from this work.

**Main Review:**

This paper looking at a very important classical online algorithms problem from a novel perspective, and got some interesting both positive and negative results. However, the theoretical part seems to be a bit lacking because the cost of the algorithm is not compared to the offline optimal cost, but rather to the optimal cost given that the facilities can only be opened in a location that has been predicted. Thus, this is equal to the offline optimal cost only if the union of the prediction sets contains all optimal locations for facilities. I think an attempt need to be made to relate the cost of the algorithm to the offline optimal algorithm, as it is the standard for online algorithms. Additionally, the presentation could also be improved. For example, Algorithm "TakeHeed" only uses the union of the advice sets and therefore some statements could be simplified. If I understand correctly, the separate advice sets are only relevant for "Robust-TakeHeed", which is not described in the main body of the paper.


Comments:
- Line 124: the variable $k$ is overloaded here. It has already been used for the number of advice sets and this could create confusion.
- Theorem 3.1: This theorem should probably be rephrased such that the advice is a single set $\mathcal{S}$ and is can just be applied on $\mathcal{S}=\cup_i \{S_i\}$ when needed, since the separate sets $S_k$ are not used.


**Time Spent Reviewing:**

4

---

> ### Author Response · Authors · 2021-08-05
> **Response to Reviewer gYHr**
>
> First, thanks for your review and comments.
>
> Your main reservation seems to be that “the theoretical part seems to be a bit lacking because the cost of the algorithm is not compared to the offline optimal cost, but rather to the optimal cost given that the facilities can only be opened in a location that has been predicted.” This appears to be an incorrect assessment of our results, however. Our end product, algorithm Wary, does compare with the optimal cost as stated in the Introduction and recalled in Corollary 3.1.1.
>
> You also suggest “to relate the cost of the algorithm to the offline optimal algorithm, as it is the standard for online algorithms”. We think that the approach you suggest would not be useful in our context, for the following reason. Since no assumptions are made on the quality of the advice, it can be totally unrelated to the online instance and, as a consequence, the optimum restricted to the advice sets can be arbitrarily far from the online optimum. So, using the latter as a benchmark cannot provide any provable guarantee. In addition, in the advice literature is standard to compare the quality of the algorithm with the quality of the advice and have worst-case guarantees if the advice has low quality (as in Corollary 3.1.1).
> Concerning the readability of the paper, we certainly agree it can be improved! We are not sure however that the specific example you mention would make things so much different, but if you have additional suggestions they would be welcome.
>
> Please let us know if further clarifications are needed.

---

### Decision · Program_Chairs · 2021-09-27

**Decision:**

Accept (Poster)

**Comment:**

The paper presents a learning-augmented algorithm for online facility location. The competitive ratio of the algorithm depends on the quality of the prediction, and the size of the advice; it is never worse than the competitive ratio of the best worst-case algorithm.
There was a substantial debate about (i) the advice model - how natural/realistic it is and (ii) the approximation guarantees, esp. the possible discrepancy between OPT and OPT(P,S). Overall, however, the reviewers felt that these issues do not significantly reduce the value of the paper.